# TS-Reasoner: Aligning Time Series Foundation Models with LLM Reasoning

## Abstract

Time series reasoning is crucial to decision-making in diverse domains, including finance, energy, and scientific discovery. While existing time series foundation models (TSFMs) can capture low-level dynamic patterns and provide accurate forecasting, further analysis usually requires additional background knowledge and sophisticated reasoning, which are lacking in most TSFMs but can be achieved through Large Language Models (LLMs). On the other hand, without expensive post-training, LLMs often struggle with the numerical understanding of time series data. Although it is intuitive to integrate the two types of models, developing effective training recipes that align the two modalities for reasoning tasks is still an open challenge. To this end, we propose TS-REASONER that aligns the latent representations of TSFMs with the textual inputs of LLMs for downstream understanding/reasoning tasks. Specifically, we propose a simple yet effective method to curate diverse, synthetic pairs of time series and textual captions for alignment training. We then develop a two-stage training recipe that applies instruction finetuning after the alignment pretraining. Unlike existing works that train an LLM to take time series as inputs, we leverage a pretrained TSFM and freeze it during training. Experiments on several benchmarks demonstrate that TS-REASONER not only outperforms a wide range of open-source LLMs, Vision Language Models (VLMs), and Time Series LLMs of comparable scale, but also achieves this with remarkable data efficiency, e.g., using less than half the training data.

## 1 Introduction

Time series analysis has long been fundamental to various real-world applications in finance, energy, weather, traffic, and other domains (Prakarsha & Sharma, 2022; Xu et al., 2023; Nie et al., 2024). Its ability to model dynamics and predict future states based on historical data makes it an indispensable tool for decision-making and strategic planning. While numerical attributes form the bedrock of time series analysis, human decision-making is often complemented by rich prior knowledge and qualitative contextual information, including news articles, social media trends, and expert assessments. This gap prevents analytical models from achieving a deeper, more contextualized understanding of the events and dynamics driving the numerical data. By enabling machines to understand both contextual information and numerical time series patterns, we can empower them as automated systems that assist humans in gaining deeper insights into complex phenomena.

Recent advances in Time Series Foundation Models (TSFMs) have significantly enhanced the understanding of time series data through large-scale pretraining. These models are capable of generalizing across various time series tasks and domains. Although TSFMs (Goswami et al., 2024; Das et al., 2024) demonstrate strong modeling capabilities, most are pre-trained exclusively on unimodal numerical time series and cannot therefore comprehend or integrate textual information. On the other hand, Large Language Models (LLMs) and Vision Language Models (VLMs) can take texts and images as input context, and have demonstrated remarkable reasoning and problem-solving abilities across various tasks (Wei et al., 2022; Yao et al., 2023; Hao et al., 2023; Yu et al., 2024; Ho et al., 2025), sparking interest in transferring their capabilities to time series analysis. Some studies (Gruver et al., 2023; Liu et al., 2024c; Jia et al., 2024) transform numerical time series into string form and perform time series forecasting on LLMs by prompting them with the strings. However, despite their strong reasoning abilities, LLMs struggle to capture temporal dependencies due to their inherent lack of

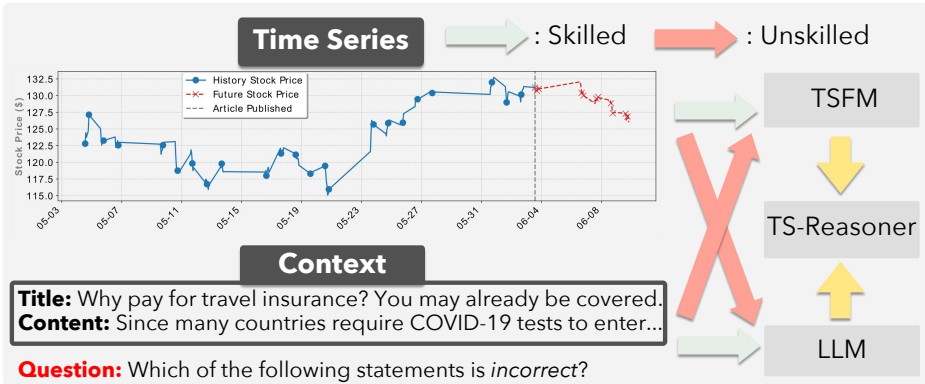

Figure 1: Time series forecasting vs. reasoning. The time series reasoning task requires both contextual reasoning (e.g., news) by LLMs and numerical understanding by TSFM.

temporal understanding (Fons et al., 2024; Merrill et al., 2024) and limited ability to interpret numerical values. These limitations hinder their understanding of time series data. As shown in Figure 1, TSFM and LLM have complementary strengths; the former specializes in temporal understanding, while the latter excels at text understanding. To combine the complementary strengths of TSFM and LLM while overcoming their respective limitations, we propose TS-REASONER, a Time Series Large Language Model (TSLLM) designed to enhance time series reasoning by aligning a TSFM with an LLM. Specifically, we first employ the TSFM to extract rich temporal representations from numerical time series data. To effectively incorporate this temporal information into the LLM, TS-REASONER introduces a TS-to-Text adapter, which projects the TSFM-extracted temporal features into the LLM's input embedding space. This enables seamless integration of the TSFM's temporal understanding with the LLM's powerful linguistic and reasoning capabilities. Our training framework consists of two stages: pretraining and fine-tuning. In the pretraining stage, we finetune TS-REASONER to produce textual captions of input time series and achieve a fundamental alignment. To this end, we propose a simple yet effective prompting strategy to curate high-quality captions for diverse time series data using advanced LLMs/VLMs. In the fine-tuning stage, we further enhance the model's reasoning abilities through instruction tuning, ensuring robust performance in downstream tasks.

Our work makes unique contributions to a recent line of research combining TSFMs and LLMs. First, our formulation sets up the connection between LLMs and TSFMs, facilitating time series reasoning through the integration of rich contextual information and LLM reasoning. Second, we address a critical data bottleneck by a simple yet effective time series captioning method, which diversifies the training data for aligning LLMs and TSFMs. Finally, we offer new empirical insights into the strengths and limitations of existing approaches.

We evaluate the understanding and reasoning capabilities of our approach on two

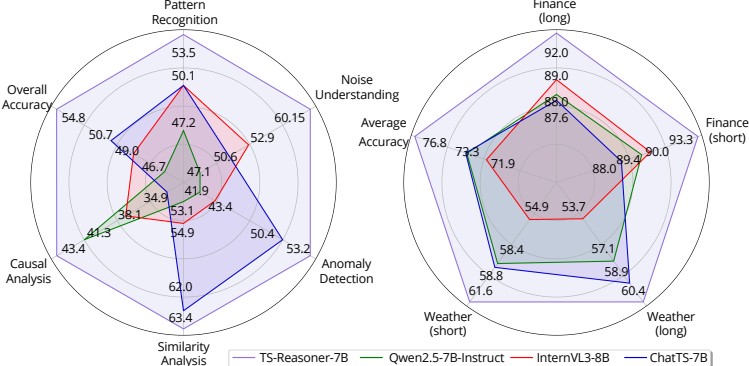

Figure 2: Results on time series understanding and reasoning benchmarks. TS-REASONER demonstrates a consistent advantage over the prevailing LLMs, VLMs, and TSLLMs.

standard benchmarks: TimeSeriesExam (Cai et al., 2024a) and MTBench (Chen et al., 2025). TS-REASONER significantly outperforms a wide range of baseline models, including LLMs, VLMs, and the TSLLMs, as shown in Figure 2. Finally, comprehensive analyses, including extensive ablation studies, validate the effectiveness of our key designs and establish the superiority of TS-REASONER in generalization performance, scalability, and training data efficiency.

## 2 Related Work

**LLMs for Time Series.** LLMs have recently garnered significant interest in time series analysis. Traditional time series forecasting relies on statistical models (RB, 1990) or data-driven neural networks (Liu et al., 2021; Lim et al., 2021; Wu et al., 2021; Zhou et al., 2022; Li et al., 2023b; 2024b) for tasks like weather and stock prediction. Recent efforts explore LLMs for this task, with some designing prompts to elicit forecasting abilities (Cao et al., 2023; Chuang* et al., 2024). Others focus on enabling LLMs to understand time series data by converting it into textual sequences or aligning its embeddings with language model embeddings via prompting or semantic information (Jin et al., 2023; Sun et al., 2023; Pan et al., 2024). In addition, multimodal vision-based LLMs are being investigated for time series prediction (Chen et al., 2024c; Zhong et al., 2025). Though LLMs exhibit non-trivial performance on some forecasting tasks, (Merrill et al., 2024) indicate that LLMs struggle to reason about time series. To tackle this challenge, several works (Chow et al., 2024; Zhang et al., 2025a; Xie et al., 2024) propose to enable LLMs to understand the time series with context. TS-Reasoner lies in this direction, distinguishing itself by employing a pre-trained Time Series Foundation Model to ground the LLM's reasoning in robust temporal features. See Appendix C for a comparison with representative architectures that leverage LLMs/VLMs for time series.

**Modality Alignment.** Modality alignment methods are widely studied in the multimodal domain (Li et al., 2022; Lai et al., 2024; Li et al., 2023a; Liu et al., 2024b). Inspired by the success of multimodal alignment, recent works treat time series as another modality and align it to the LLM (Xie et al., 2024; Zhang et al., 2025a). Though they achieve a certain degree of time series understanding, they focus on narrow domains (e.g., electricity) and tasks (e.g., time series understanding), and train time series encoders from scratch. In contrast, we adapt the successful training paradigm in VLMs, identify and address the key challenges (e.g., integration of characteristics of time series into LLMs, and the shortage of time series-text pairs) faced in applying this paradigm to the unique modality of time series, exploring pre-trained time series foundation models to exploit rich time series knowledge.

**Time Series Foundation Models.** Recent advancements in pre-training methods are significantly contributing to the development of foundation models for time series analysis. Early efforts, such as TST (Zerveas et al., 2021) and PatchTST (Nie et al., 2022), applied BERT-like masked pretraining techniques, focusing on point-level and patch-level masking, respectively. A separate line of work, exemplified by models like TimesFM (Das et al., 2024), Timer (Liu et al., 2024e), TTMS (Ekambaram et al., 2024), Chronos (Ansari et al., 2024), Time-MoE (Shi et al., 2024), Moirai (Liu et al., 2024d), CoRA (Qin et al., 2025), TimesBERT (Zhang et al., 2025b), and Sundial (Liu et al., 2025) demonstrates the advantages of large-scale pre-training for improving forecasting performance. Exploring diverse pre-training objectives, MOMENT (Goswami et al., 2024) leverages a T5 encoder to achieve strong downstream multi-task capabilities. ChronoSteer (Wang et al., 2025a) also explores the alignment between TSFMs and LLMs, yet it leverages the LLM's revisions to enhance TSFM's forecasting capability.

## 3 TS-Reasoner for Temporal Reasoning

### 3.1 Problem Formulation

We start by defining time series reasoning. Given a natural language context $\mathcal{X}$, which may encode background domain knowledge or prompt instructions, and a corresponding set of time series $\mathcal{S} = \{\mathcal{T}_1, \ldots, \mathcal{T}_K\}$, the model produces an output sequence $\mathcal{V}$. To explicitly model the reasoning capability, we formulate $\mathcal{V}$ as a composition of a reasoning path $\mathcal{R}$ and a final answer $\mathcal{A}$, denoted as $\mathcal{V} = [\mathcal{R}; \mathcal{A}]$. Here, $\mathcal{R}$ consists of tokens providing intermediate explanations for problem-solving, and $\mathcal{A}$ contains the tokens for the final conclusion.

The model defines a probability distribution over the output sequence $\mathcal{V}$, conditioned on inputs $\mathcal{X}$ and $\mathcal{S}$. This generation process can be factorized as:

$$P(\mathcal{V}|\mathcal{X}, \mathcal{S}) = \underbrace{P(\mathcal{A}|\mathcal{R}, \mathcal{X}, \mathcal{S})}_{\text{Answer Generation}} \cdot \underbrace{P(\mathcal{R}|\mathcal{X}, \mathcal{S})}_{\text{Reasoning Process}} \tag{1}$$

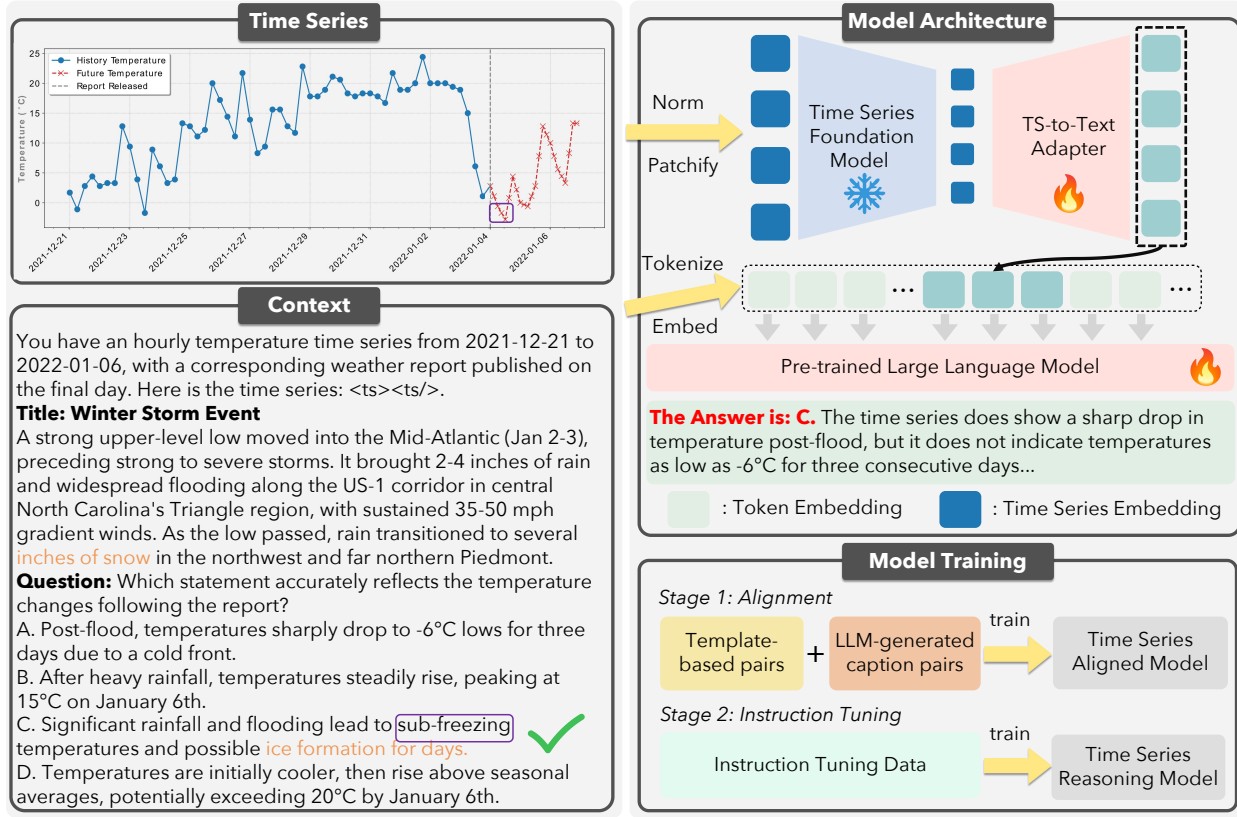

Figure 3: Overview of TS-REASONER architecture and training pipeline. To perform reasoning, a time series is first encoded by a pretrained Time Series Foundation Model (TSFM). Its output features are then projected into the LLM's input embedding space by a trainable TS-to-Text Adapter and subsequently processed by the LLM. The model is trained in two stages: (1) a pretraining stage that aligns the TSFM outputs with the LLM inputs using both template-based (code-synthesized) and LLM-generated captions, as described in §3.4, and (2) an instruction-tuning stage to improve complex reasoning capabilities.

where $P(\mathcal{R}|\mathcal{X}, \mathcal{S})$ indicates the generation of the reasoning path that explains the logical derivation based on the multimodal context, and $P(\mathcal{A}|\mathcal{R}, \mathcal{X}, \mathcal{S})$ represents the generation of the final answer conditioned on the generated reasoning path.

## 3.2 Overview

As illustrated in Figure 3, TS-REASONER is composed of (1) a pretrained TSFM that encodes normalized, non-overlapping patches of input time series into compact embeddings; (2) a pretrained LLM, and (3) a TS-to-Text adapter that projects the TSFM's output embedding to the input space of the LLM. The LLM concatenates the sequence of projected time series features with the sequence of embeddings for input text tokens, with the former demarcated by special tokens "$\langle ts \rangle \langle ts / \rangle$". The training of TS-REASONER consists of two stages: (1) a pretraining stage to align time series features from the TSFM with the LLM, using time series caption data synthesized by an advanced LLM/VLM, and (2) an instruction tuning stage to enhance complex reasoning capabilities on downstream tasks.

## 3.3 Model Architecture

Given a natural language context $\mathcal{X}$ and a corresponding set of time series $\mathcal{S} = \{\mathcal{T}_1, \mathcal{T}_2, \ldots, \mathcal{T}_K\}$, we first project both into a shared embedding space. Specifically, for each time series $\mathcal{T}_i \in \mathbb{R}^{L_i}$, where $L_i$ is the length of the series, we first apply a value-preserved normalization: we subtract the mean of the series $\mu_i$, and further divide the centered series by a scale factor per-series $s_i$ when its magnitude exceeds a fixed stability range.

This preprocessing step ensures that the model is robust to shifts and scales in the input data. Meanwhile, to preserve the absolute magnitude information, the offset $\mu_i$ and scale $s_i$ are inserted into the corresponding time-series placeholder in $\mathcal{X}$ as plain-text numerical tokens, from which the original values can be exactly recovered. Subsequently, we partition the normalized time series into a sequence of non-overlapping patches, each of a fixed length $P$. This patching strategy yields a sequence of $N_i = \lfloor L_i/P \rfloor$ patches, transforming the time series into a tensor $\mathcal{T}_i^p \in \mathbb{R}^{N_i \times P}$. These patches are then encoded using the TSFM, which acts as our time series feature extractor. The TSFM processes the sequence of patches and produces a sequence of embedding vectors:

$$\mathcal{Z}_i^T = \text{TSFM}(\mathcal{T}_i^p) \in \mathbb{R}^{N_i \times d_{\text{ts}}}, \tag{2}$$

where $d_{\text{ts}}$ denotes the dimension of the time series embeddings. Concurrently, the natural language context $\mathcal{X}$ is tokenized and fed into the pre-trained LLM's embedding layer. This process converts the textual input into a sequence of contextualized token embeddings:

$$\mathcal{Z}^L = \text{LLM}_{\text{embed}}(\mathcal{X}) \in \mathbb{R}^{M \times d_{\text{text}}}, \tag{3}$$

where $M$ is the number of tokens in the instruction, and $d_{\text{text}}$ is the dimensionality of the LLM's hidden states. To align the dimension and semantics of embeddings between LLM and TSFM, we use a single layer multilayer perceptron (MLP) as a TS-to-Text Adapter to transform the time series embedding into the text embedding space:

$$\mathcal{H}_i^T = \text{MLP}(\mathcal{Z}_i^T) \in \mathbb{R}^{N_i \times d_{\text{text}}}, \tag{4}$$

To form a unified input sequence for the LLM that accommodates multiple time series, we structure the natural language instruction $\mathcal{X}$ to include $K$ indicators, $\{K \cdot \langle\text{ts}\rangle\langle\text{ts}/\rangle\}$. The $i$-th placeholder $\langle\text{ts}\rangle\langle\text{ts}/\rangle$ marks the insertion point for the corresponding $i$-th time series $\mathcal{T}_i$.

Let $\{\mathcal{H}_i^T \in \mathbb{R}^{N_i \times d_{\text{text}}}\}_{i=1}^K$ be the set of projected time series embeddings, The final input sequence $\mathcal{H}$ is constructed by sequentially inserting the embedding to each $\langle\text{ts}\rangle\langle\text{ts}/\rangle$ with its corresponding time series embedding sequence $\mathcal{H}_i^T$. This substitution process results in a composite sequence where language and time series representations are interleaved. The total length of this fused sequence is $M + \sum_{i=1}^K N_i$. The final tensor fed to the LLM's transformer layers is therefore: $\mathcal{H} \in \mathbb{R}^{(M + \sum_{i=1}^K N_i) \times d_{\text{text}}}$. This strategy enables the LLM to process multiple, arbitrarily placed time series within a single, coherent context and capture complex inter-series and text-series dependencies. After the combination, the input embedding $\mathcal{H}$ is fed to the LLM to produce the final prediction $\mathcal{Y}$.

### 3.4 Training Recipe

Our training process consists of two sequential stages: the first stage aligns time series data with the LLM to establish a foundational understanding of temporal-textual relationships, while the second stage refines the LLM's reasoning capabilities to interpret and analyze these aligned representations. Throughout both stages, we keep the parameters of the TSFM frozen to preserve its pretrained temporal knowledge, while allowing the LLM's parameters to remain trainable, ensuring adaptive learning without compromising the integrity of the encoded time series features.

**Stage 1: Pre-training for Language-Timeseries Alignment.** This stage aligns temporal data with textual information. We initially leverage synthesized data from (Xie et al., 2024), which provides predefined templates to describe time series attributes. Although these template-based data offer accurate numerical information, their focus on specific time series patterns limits diversity, and the caption structure is monotonous. This lack of diversity can lead to overfitting to the templates, encouraging the model to learn shallow patterns and resulting in poor generalization (Dong et al., 2025; Choi et al., 2024). To alleviate this problem, we draw inspiration from captioning techniques in multimodal LLMs (Chen et al., 2024a). We synthesize comprehensive captions using advanced models (e.g., GPT-4.1) to enrich our alignment data. Specifically, we collect time series from two sources: (Merrill et al., 2024), which includes contextual information, and synthetic data from Chronos (Ansari et al., 2024), which provides pure numerical time series.

**Attribute-aware Captioning.** Caption generation has been extensively investigated in visual domains (Cheng et al., 2023; 2025; Chen et al., 2024b), playing a crucial role in multimodal alignment.

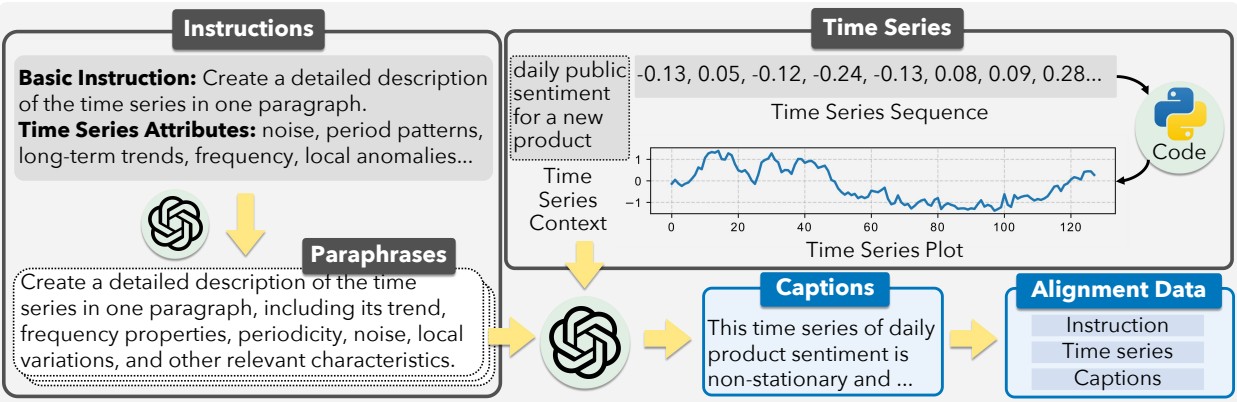

Figure 4: Workflow for our attribute-aware caption synthesis, designed to curate training data for alignment in stage 1. It enriches basic instructions with key attributes and generates diverse paraphrases, yielding the high-fidelity captions to train TS-REASONER effectively.

However, time series captioning remains largely underexplored, presenting a significant impediment to achieving comprehensive alignment. To address this gap, we introduce a straightforward approach for generating scalable time series captions, as shown in Figure 4.

Given a time series $\mathcal{T}$ with a temporal context $\mathcal{C}$, we begin by defining a fundamental captioning instruction, denoted as $\mathcal{I}_{\text{base}}$. To facilitate enhanced comprehension by LLMs, we transform the time series into an image plot via Python code, $I_{TS} = \Phi(\mathcal{T})$. As evidenced in Table 2 (Section 4.1), presenting the time series as an image to advanced LLMs (e.g., GPT-4.1) demonstrates a substantial advantage in understanding compared to providing it as a raw numerical series.

To enrich the generated captions, we first identify $G$ pertinent attributes of the time series, denoted as $\{a_1, a_2, \dots, a_G\}$ (e.g., trend, frequency, periodicity, noise, local variations). These attributes are then incorporated into the basic instruction, yielding an augmented instruction $\mathcal{I}' = \mathcal{I}_{\text{base}} \cup \{a_1, a_2, \dots, a_G\}$. To further promote caption diversity, we use the LLM to paraphrase $\mathcal{I}'$ into $R$ distinct instructions, forming a candidate set of prompts $\mathcal{P} = \{\mathcal{I}''_1, \mathcal{I}''_2, \dots, \mathcal{I}''_R\}$. For each time series $\mathcal{T}$, a prompt $\mathcal{I}''$ is uniformly sampled from this set, and the caption is generated conditioned on it and the time series visualization:

$$\text{Caption} = \text{LLM}(\mathcal{I}'', I_{TS}), \tag{5}$$

where $\mathcal{I}'' \sim \mathcal{U}(\mathcal{P})$. The prompts are shown in Figure 12 in Appendix E. We randomly sample 10K time series from each of two distinct sources: the Chronos synthetic dataset (Ansari et al., 2024), which contains purely numerical time series, and a dataset of text-attributed time series from (Merrill et al., 2024), which provides contextual backgrounds. The construction of data offers two benefits: (1) Pure time series data enables the model to build a foundational understanding of temporal patterns by focusing solely on the intrinsic characteristics of the data. (2) Context-augmented time series enhances domain-specific comprehension by linking numerical trends to real-world scenarios, thereby improving the model's ability to generalize across diverse applications.

**Stage 2: Instruction Finetuning for Time Series Reasoning.** To elevate the model's capabilities from foundational understanding to complex reasoning, we employ an instruction fine-tuning stage based on the instruction tuning dataset (Xie et al., 2024), which encompasses a wide range of Q&As and instruction-following tasks. This training equips TS-REASONER with two critical abilities: the fidelity to adhere to complex instructions and structured response formats, and the capacity for nuanced, context-driven reasoning on time series-specific queries.

# 4 Experiments

**Datasets.** To assess the capabilities of TS-REASONER, we conduct comparative experiments against various baselines on benchmarks tailored for time series reasoning. Our evaluation incorporates TimeSeriesExam (Cai

et al., 2024a), a comprehensive multiple-choice question answering dataset. TimeSeriesExam is specifically engineered to systematically evaluate a model's time series understanding and reasoning abilities across several key aspects: Pattern Recognition (PR), which addresses identifying trends, cycles, and stationarity; Noise Understanding (NU), focused on recognizing noise types such as white noise and random walks; Anomaly Detection (AD), for detecting unusual patterns; Similarity Analysis (SA), which involves comparing the shape and distribution of two time series; and Causality Analysis (CA), assessing the recognition of Granger Causality between time series. See Table 7 for more detailed descriptions. Furthermore, we evaluate on MTBench (Chen et al., 2025), a large-scale benchmark for evaluating time series reasoning in the real-world financial and weather domains, featuring questions that span both short-term (7-day) and long-term (14-day) temporal horizons, serving as a supplementary dataset since MTBench is a non-peer-reviewed preprint. Additionally, we evaluate an inductive reasoning task from (Xie et al., 2024) for open-ended reasoning ability.

**Baselines and Evaluation Metrics.** We compare our method against three types of baselines: closed-source LLMs / VLMs, open-source LLMs / VLMs, and TSLLMs. Specifically, for closed-source models, we include GPT-4o, GPT-4.1 (gpt, 2024; Achiam et al., 2023), Claude-Sonnet-3.7 (The), and DeepSeek-Chat (Liu et al., 2024a). For open-source LLMs, we evaluate Llama-3.1-8B-Instruct (Grattafiori et al., 2024), Qwen2.5-7B-Instruct (Yang et al., 2024), GLM-4-9B-Chat (GLM et al., 2024), InternLM3-8B-Instruct (Cai et al., 2024b), and Ministral-8B-Instruct (Liu et al., 2026). Time series are transformed into textual sequences of numbers for LLMs. For open-source VLMs, we compare Qwen2.5-VL-7B (Bai et al., 2025), Phi-4-Multimodal-Instruct (Abouelenin et al., 2025), Llama3-LLaVA-Next-8B (Li et al., 2024a), InternVL3-8B (Zhu et al., 2025), and MiniCPM-V-2.6 (Yao et al., 2024). Time series are transformed into plots via code for VLMs. For TSLLM models, we compare with ChatTime-7B (Wang et al., 2025b), ChatTS-14B (Xie et al., 2024), and we use the official training data and code to fine-tune a 7B model for a fair comparison. As all benchmarks are multiple-choice Q&As, we use accuracy as the evaluation metric. Reported accuracies are averaged over three inference runs of a single trained checkpoint using different random seeds. We further apply McNemar's test on the paired test-set predictions to assess the statistical significance of the improvements, and base our robustness claims on this test-set significance.

**Implementation Details.** TS-REASONER uses the Qwen2.5-7B-Instruct as the LLM backbone across all the experiments with an embedding dimension of 3584, and uses the TimesFM-1.0-200M (Das et al., 2024) as our backbone TSFM with an embedding dimension of 1280. The TS-to-Text adapter is a single-layer MLP with a GELU activation, which projects the 1280-dimensional TSFM output features into the 3584-dimensional hidden space of the LLM. Parameters of the LLM are fine-tuned while those of the TSFM remain frozen during training. The detailed derivation of these time series embeddings from TimesFM can be found in Appendix A. We use $R = 20$ different instructions to generate captions, and integrate $G = 5$ different attributes into the instructions. The training processes were conducted on $8 \times$ L40s GPUs, where

Table 1: Training details of TS-REASONER.

|  | **Stage-1** | **Stage-2** |
|---|---|---|
| Patch Size | 32 | 32 |
| Dataset | Captions | Instructions |
| #Samples | 120K | 30K |
| TSFM | TimesFM-1.0-200M | |
| LLM Backbone | Qwen2.5-7B-Instruct | |
| Trainable Params | 7.3B | 7.3B |
| Batch Size | 64 | 32 |
| Learning Rate | $1 \times 10^{-5}$ | $2 \times 10^{-5}$ |
| Epoch | 1 | 2 |

stage 1 and 2 consume about 20 and 5 hours, respectively. In stage 1, training data is composed of 100K template-based pairs and 20K LLM-generated caption pairs. In stage 2, we employ instruction tuning data from (Xie et al., 2024). Comprehensive training parameters are further detailed in Table 1.

## 4.1 Main Results

Table 2 presents the performance of all models on the two benchmarks. The best results are bolded, and the second-best results are underlined. Based on the results, we have the following key observations:

**(i) TS-Reasoner achieves superior overall performance on all benchmarks among models of the same size.** Specifically, TS-REASONER demonstrates superior performance, surpassing the best-performing LLM by 8.17% overall, the best VLM by 5.82% overall, and the TSLLM by 4.11% overall on the TimeSeriesExam benchmark. Compared to the backbone model, TS-REASONER improves on our backbone LLM performance by a substantial 17.50%. TS-REASONER also outperforms the best baseline on MTBench

Table 2: Performance of LLMs, VLMs, TSLLMs, and proprietary models on time series understanding and reasoning benchmarks. $\pm$ values are standard deviations over three inference runs of a single checkpoint. * marks TS-Reasoner improvements that are statistically significant under McNemar's test on the test set ($p < 0.05$). Our baselines also include ChatTS-14B, which uses a larger base model.

| Model | TimeSeriesExam (Cai et al., 2024a) | | | | | | MTBench (Chen et al., 2025) | | | |
|---|---|---|---|---|---|---|---|---|---|---|
| | PR | NU | AD | SA | CA | OA | Finance (long) | Finance (short) | Weather (long) | Weather (short) |
| *Proprietary models* | | | | | | | | | | |
| DeepSeek-Chat | 65.23 | 55.17 | 52.71 | 63.71 | 42.86 | 59.89 | 89.15 | 90.02 | 59.75 | 58.76 |
| DeepSeek-R1 | 74.66 | 63.22 | 63.56 | 65.49 | 41.27 | 67.36 | 65.31 | 60.69 | 49.45 | 46.36 |
| Claude-Sonnet-3.7 | 62.26 | 55.17 | 48.06 | 72.57 | 50.79 | 59.63 | 84.11 | 88.56 | 51.24 | 47.91 |
| GPT-4o | 59.03 | 55.17 | 53.49 | 62.83 | 31.75 | 55.96 | 84.30 | 82.69 | 48.07 | 48.22 |
| GPT-4o (vision) | 67.12 | 62.07 | 62.79 | 64.60 | 26.98 | 62.12 | 84.11 | 80.65 | 46.43 | 48.53 |
| GPT-4.1 (vision) | 69.81 | 68.97 | 68.22 | 75.22 | 41.27 | 67.89 | 93.41 | 91.45 | 56.04 | 55.35 |
| *Open-source Large Language Models* | | | | | | | | | | |
| Llama-3.1-8B-Instruct | 37.73 | 37.93 | 30.23 | 36.28 | 28.57 | 35.52 | 63.37 | 35.52 | 40.25 | 40.00 |
| Qwen2.5-7B-Instruct | 47.17 | 47.13 | 41.86 | 53.10 | 41.27 | 46.66 | 87.98 | 89.41 | 57.14 | 58.44 |
| GLM-4-9B-chat | 41.78 | 39.08 | 37.21 | 47.79 | 38.09 | 41.28 | 71.31 | 77.19 | 50.27 | 50.85 |
| InternLM3-8B-Instruct | 43.93 | 51.72 | 26.35 | 52.21 | 34.92 | 42.33 | 71.70 | 71.08 | 45.05 | 46.67 |
| Ministral-8B-Instruct | 43.13 | 37.93 | 39.53 | 44.25 | 36.51 | 41.55 | 46.32 | 50.71 | 39.15 | 40.93 |
| *Open-source Vision Language Models* | | | | | | | | | | |
| Qwen2.5-VL-7B-Instruct | 25.34 | 32.18 | 19.38 | 42.48 | 12.70 | 26.61 | 81.98 | 86.35 | 52.06 | 46.82 |
| Phi-4-Multimodal-Instruct | 36.39 | 34.48 | 30.23 | 38.94 | 14.28 | 33.68 | 70.35 | 74.54 | 48.35 | 49.77 |
| Llama3-LLaVA-Next-8B | 31.27 | 35.63 | 29.46 | 30.09 | 38.09 | 31.85 | 52.14 | 51.50 | 47.53 | 47.29 |
| InternVL3-8B | 50.13 | 52.87 | 43.41 | 54.87 | 38.09 | 49.01 | 88.95 | 90.00 | 53.71 | 54.88 |
| MiniCPM-V-2.6 | 29.11 | 39.08 | 27.13 | 51.33 | 31.75 | 33.42 | 81.78 | 83.09 | 48.63 | 45.12 |
| *Time Series Large Language Models* | | | | | | | | | | |
| ChatTime-7B | 42.85 | 49.42 | 35.65 | 44.24 | 34.92 | 41.94 | 25.97 | 28.10 | 47.80 | 42.79 |
| ChatTS -7B | 50.13 | 50.57 | 50.38 | 61.95 | 34.92 | 50.72 | 87.60 | 88.01 | 58.92 | 58.75 |
| ChatTS -14B* | 59.30 | 54.02 | 51.16 | 62.83 | 41.27 | 56.36 | 89.22 | 91.22 | 59.61 | 59.22 |
| TS-REASONER-7B (ours) | $53.46^*_{\pm1.58}$ | $60.15^*_{\pm3.51}$ | $53.23^*_{\pm1.61}$ | $63.42^*_{\pm1.02}$ | $43.39^*_{\pm1.98}$ | $54.83^*_{\pm0.98}$ | $92.00^*_{\pm1.74}$ | $93.28^*_{\pm1.28}$ | $60.44^*_{\pm0.14}$ | $61.55^*_{\pm0.31}$ |
| $\Delta$ Over Best 7B | +3.33 | +7.28 | +2.85 | +1.47 | +2.12 | +4.11 | +3.05 | +3.28 | +1.52 | +2.80 |

by ~2–3%. In addition, TS-REASONER also performs competitively against ChatTS-14B, which has a larger base model. The notable improvement demonstrates the effectiveness of our model in various time series reasoning scenarios by introducing the temporal information of TSFM for the LLM.

(ii) **TS-Reasoner shows consistent gains across time series reasoning subtasks.** TS-REASONER consistently leads across reasoning tasks, with notable improvements over the second-best baseline in multiple subtasks like *Pattern Recognition* (3.33%), *Noise Understanding* (7.28%). It also gains ~2% in financial and weather reasoning. Such improvements stem from robust alignment and the enhanced ability to reason over numerical patterns within textual contexts.

Among all categories, Pattern Recognition shows the largest gap to GPT-4.1. We attribute this primarily to the general multi-step reasoning capacity gap between our 7B backbone and a frontier-scale model. This suggests that while TS-Reasoner improves the model's time series understanding, its overall performance on tasks that demand both temporal understanding and complex multi-step reasoning remains bounded by the reasoning capacity of the base LLM. A promising direction for future work is to further strengthen this reasoning ability on top of the current model to narrow the gap to frontier models.

## 4.2  Analysis of Data Scaling and Efficiency

Figure 5 presents our data scaling analysis on the TimeSeriesExam benchmark. TS-REASONER demonstrates remarkable data efficiency compared to the ChatTS-7B baseline. For the alignment stage, TS-REASONER achieves superior overall accuracy using just 60K samples, less than half the data required by the baseline. This efficiency is even more stark in the instruction tuning stage, where 10K samples suffice to outperform ChatTS-7B. This significant reduction in data dependency stems from our pre-trained TSFM and effective alignment, which equips the LLM with a robust temporal foundation. Consequently, TS-REASONER develops advanced reasoning capabilities with a substantially smaller amount of data, marking a key advantage for practical deployments where labeled data is scarce.

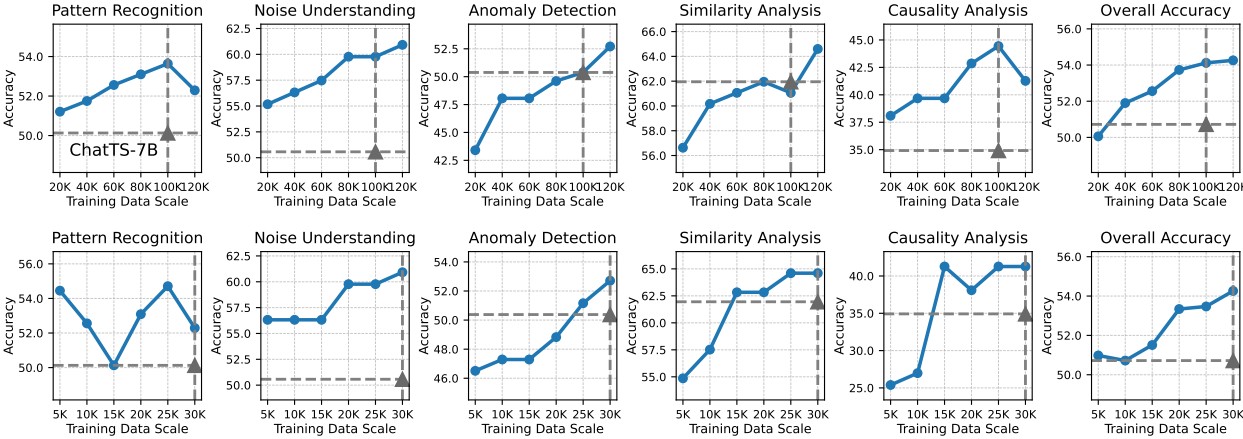

Figure 5: Data scaling and efficiency of TS-REASONER. The top (bottom) row illustrates how the performance of TS-REASONER varies when increasing the training data for alignment (instruction tuning). The columns correspond to sub-tasks in TimeSeriesExam. ChatTS-7B (Xie et al., 2024) is included for reference, denoted by the gray triangle.

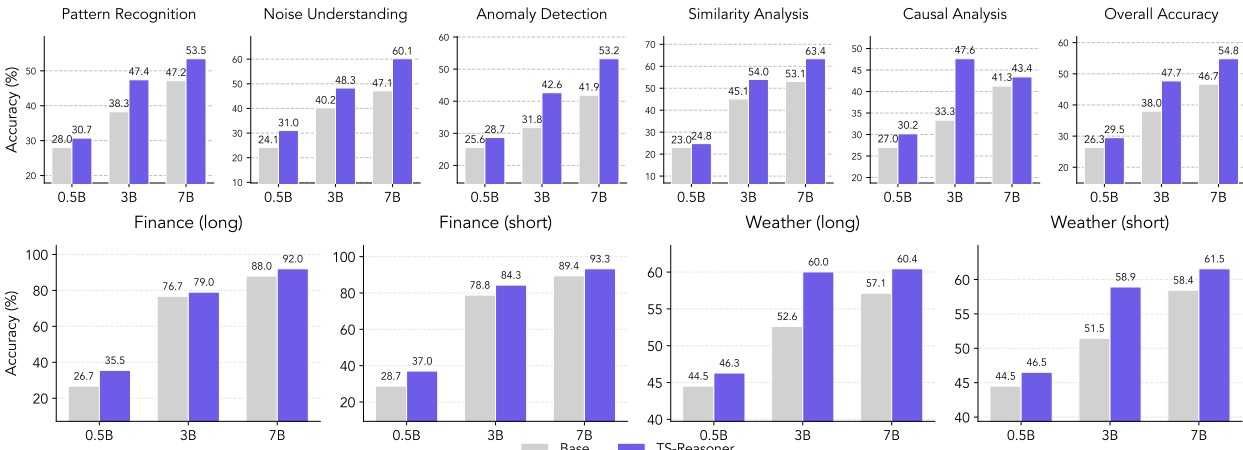

Figure 6: Performance of TS-REASONER and its associated LLM backbones (Qwen2.5 series). The top row and bottom row report the performance on TimeSeriesExam and MTBench, respectively.

### 4.3 Choices of Captioning Model for Alignment

The quality of the generated captions is a critical factor in the efficacy of our time-series-language alignment. To validate this, we trained TS-REASONER using three distinct sets of captioning data, each generated by a model with varying capabilities: the state-of-the-art GPT-4.1, and two VLMs, InternVL3-8B and Qwen2.5-VL-7B-Instruct. As illustrated in Figure 7, the results demonstrate that the performance of TS-REASONER is directly correlated with the fidelity of the captioning model. A distinct performance hierarchy emerges across both benchmarks: the model trained on GPT-4.1 captions consistently outperforms the one trained on InternVL3-8B captions, which in turn surpasses the one trained on Qwen2.5-VL-7B-Instruct captions. The higher performance gain from GPT-4.1 is attributed to its advanced capability in time series understanding. It is not surprising that the captions generated by InternVL3-8B achieve higher performance than Qwen2.5-VL-7B-Instruct, as its better time series understanding capability is shown in Table 2.

### 4.4 Choices of TSFM and LLM in TS-Reasoner

**Different choices of TSFMs.** To investigate the performance of TS-REASONER with different TSFMs, we replaced TimesFM (200M) with MOMENT-1-base (200M) and Chronos-base (200M), TSFMs of the same size, and re-evaluated its performance on the TimeSeriesExam benchmark. Results presented in

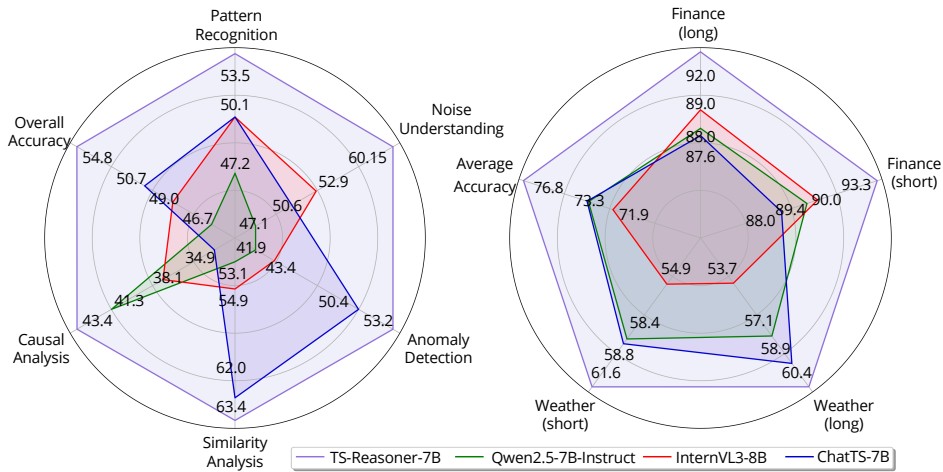

Figure 7: Comparison of multimodal LLMs used to generate time series captions for training TS-REASONER. **Left**: performance on TimeSeriesExam. **Right**: Performance on MTBench.

Table 3 reveal a substantial performance degradation when using MOMENT and Chronos, with overall accuracy falling from 54.83% to 45.74% and 53.21%, respectively. These results align with established forecasting benchmarks (Shi et al., 2024; Mulayim et al., 2024) where TimesFM demonstrates higher fidelity.

This suggests that TimesFM provides better time series representations, enabling TS-REASONER to understand and reason about time series better.

**Different choices of LLMs.** To investigate the scalability and robustness of our approach with different LLM backbones, we evaluate TS-REASONER across three distinct sizes of the Qwen2.5-Instruct backbone: 0.5B, 3B, and 7B. The results,

Table 3: Comparison of TS-REASONER using different TSFMs on the TimeSeriesExam benchmark.

| Model | PR | NU | AD | SA | CA | OA |
|---|---|---|---|---|---|---|
| MOMENT | 46.90 | 47.13 | 41.86 | 54.87 | 28.57 | 45.74 |
| Chronos | 51.75 | 59.77 | 51.93 | 63.71 | 36.51 | 53.21 |
| TimesFM | **53.46** | **60.15** | **53.23** | 63.42 | **43.39** | **54.83** |

shown in Figure 6, confirm that TS-REASONER is both highly effective and performs robustly. We observe a clear positive scaling law for both TS-REASONER and a baseline. More importantly, TS-REASONER maintains a consistent and significant lead across all models, with Overall Accuracy improvements of +3.15% (29.49% vs. 26.34%), +9.70% (47.71% vs. 38.01%), and +8.18% (54.83% vs. 46.65%) for the 0.5B, 3B, and 7B models, respectively. This demonstrates that our approach performs robustly across different LLM backbones for complex time series reasoning. In addition, the performance gain from incorporating the TSFM is markedly smaller at 0.5B than at 3B and 7B, indicating that the TSFM's contribution grows with model scale. We speculate that this is because smaller models have insufficient capacity and limited base reasoning ability, which constrains how well they can align with and exploit the TSFM features. In contrast, larger models possess a higher-dimensional, semantically richer representation space and stronger context-integration and multi-step reasoning abilities, allowing them to more fully align, interpret, and exploit the injected temporal features, thereby converting the fine-grained temporal information into substantial downstream gains.

## 4.5 Comparison between Textual and Visual Time Series for Captioning

To quantitatively assess the impact of different time series representations on captioning performance, we employed GPT-4.1 to synthesize training pairs based on either raw numerical sequences (textual) or visual plots (visual). As illustrated in Figure 8, TS-REASONER trained on visually-derived captions consistently outperforms its text-derived counterpart across most tasks. This indicates that the compact and holistic nature of visual plots enables LLMs to generate higher-fidelity captions, thereby facilitating more effective model alignment. To supplement these quantitative results, we provide a qualitative case analysis in Appendix B, which further demonstrates that visual time-series representations lead to a more accurate pattern recognition.

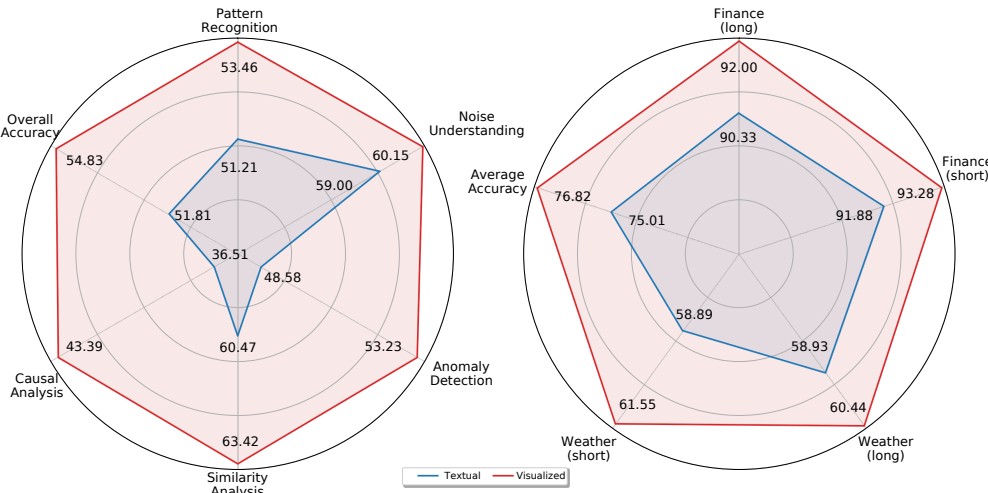

Figure 8: Results of TS-Reasoner on TimeSeriesExam (left) and MTBench (right) using textual and visualized time series for captioning as training data.

## 4.6 Caption Diversity Analysis

A critical limitation of synthetic datasets is the risk of models learning spurious correlations from similar templates. To mitigate this, our attribute-aware generation process is designed to produce captions that are lexically diverse. To quantitatively validate the richness of our approach, we compare it against the template-based method. We evaluate both lexical diversity using the Measure of Textual Lexical Diversity (MTLD) (Bestgen, 2024) and Self-BLEU-4 (Zhu et al., 2018) on a random sample of 1K captions from each dataset.

The results presented in Table 4 show that our attribute-aware captions achieve an MTLD score of 133.30, a nearly 3 times increase over the template-based score of 42.95. Furthermore, the Self-BLEU-4 score is almost halved from 0.82 to 0.45. This substantial improvement in lexical diversity confirms that our method generates a significantly more expressive and diverse set of captions, crucial for training robust and generalizable models. We provide an example in Figure 11 to compare the template-based caption and the LLM-generated caption.

Table 4: Comparison of lexical diversity between template-based pairs and LLM-generated pairs.

| Metrics | MTLD ↑ | Self-BLEU-4 ↓ |
|---|---|---|
| Template-based pairs | 42.95 | 0.82 |
| LLM-generated pairs | **133.30** | **0.45** |

To ensure comprehensive data coverage, we curated time series with context from a wide range of domains. The distribution of these domains is visualized in Figure 9.

Table 5: Results on time series inductive reasoning.

## 4.7 Open-ended Time Series Reasoning

To evaluate TS-Reasoner on open-ended time series reasoning tasks, we use the Inductive Reasoning dataset (Xie et al., 2024), which requires the model to summarize the underlying physical principles reflected by uni/multivariate time series. Following (Xie et al., 2024), we adopt RAGAS (Es et al., 2024), an LLM-based keyword-matching evaluation framework. To verify its reliability, we randomly sample 100 cases for manual inspection and find 98% agreement between the LLM evaluator and human judgments, confirming the trustworthiness of this evaluation protocol.

| Model | Accuracy |
|---|---|
| GPT-4o-mini | 33.30 |
| GPT-4o | 33.60 |
| GPT-4o-mini (vision) | 32.30 |
| GPT-4o (vision) | 32.20 |
| Qwen2.5-14B | 18.40 |
| ChatTS-7B | 50.20 |
| ChatTS-14B | 51.80 |
| TS-Reasoner-7B | **54.70** |

**Results.** As shown in Table 5, TS-Reasoner-7B outperforms all baselines, including GPT-4o and ChatTS-7B by 21.1% and 4.5%, respectively. This substantial performance gap highlights the effectiveness of our approach in capturing complex temporal dynamics and performing reasoning on open-ended tasks.

Table 6: Ablation study results of different components in TS-Reasoner. * marks improvements that are statistically significant under McNemar's test ($p < 0.05$) compared to the removal of TSFM.

| Model | TimeSeriesExam (Cai et al., 2024a) | | | | | | MTBench (Chen et al., 2025) | | | |
|---|---|---|---|---|---|---|---|---|---|---|
| | PR | NU | AD | SA | CA | OA | Finance (long) | Finance (short) | Weather (long) | Weather (short) |
| TS-Reasoner-7B | **53.46*** | **60.15*** | **53.23*** | 63.42 | **43.39*** | **54.83*** | **92.00*** | **93.28*** | **60.44*** | **61.55*** |
| *Ablation on Training Data* | | | | | | | | | | |
| - LLM-caption | 51.21 | 56.32 | 52.71 | 56.54 | 36.51 | 51.25 | 88.67 | 89.40 | 58.24 | 59.69 |
| - Attributes | 52.02 | 57.47 | 48.83 | 62.83 | 39.68 | 52.69 | 89.71 | 89.20 | 57.28 | 59.07 |
| *Ablation on Training Stages* | | | | | | | | | | |
| - Stage 1 | 47.98 | 54.02 | 37.98 | 57.52 | 30.16 | 46.92 | 80.24 | 83.71 | 52.88 | 55.34 |
| - Stage 2 | 33.42 | 28.73 | 13.95 | 25.67 | 1.59 | 25.81 | 88.07 | 86.76 | 56.86 | 58.60 |
| *Ablation on Model Architecture* | | | | | | | | | | |
| - TSFM | 51.48 | 52.87 | 51.16 | 63.71 | 38.09 | 51.76 | 89.43 | 89.70 | 58.65 | 60.62 |
| - Freeze | 51.48 | 56.32 | 53.48 | 60.18 | 41.27 | 52.81 | 89.15 | 91.02 | 58.24 | 60.15 |
| TS-as-text | 48.25 | 55.17 | 42.64 | 53.98 | 50.45 | 49.67 | 88.95 | 90.43 | 58.10 | 59.38 |

## 4.8 Ablation Studies

To further demonstrate the effectiveness of TS-Reasoner, we conduct ablation studies to analyze the impact of individual components. Table 6 summarizes our component-wise ablations from both training and model architecture perspectives:

(1) **Attribute-aware captioning is critical for robust language-timeseries alignment.** Removing the captioning data or stripping attributes from instructions degrades performance by up to 3.01% on TimeSeriesExam and ∼2% on MTBench. This confirms that fine-grained linguistic descriptions are essential for capturing nuanced temporal patterns.

(2) **Absence of any training stage significantly harms the performance.** Removing Stage 1 (Alignment) primarily impacts MTBench's cross-modal reasoning tasks, while omitting Stage 2 (Instruction Tuning) causes a 28.45% drop on TimeSeriesExam. This suggests that while alignment grounds the model, instruction tuning is essential for activating the ability to follow specific analytical commands.

(3) **Pretrained TSFM is beneficial for effective time series feature extraction.** We remove the pretrained TSFM and repurpose the TS-to-Text adapter to directly project time series patches into the LLM's embedding space. As shown in Table 6, this modification leads to a performance decrease of 3.07% on the TimeSeriesExam benchmark and 2.22% on MTBench. We further replace time series embedding modules and use textual time series to train the base model with the same pipeline, which shows a 5.16% decrease on the TimeSeriesExam benchmark and 2.61% on MTBench. These results underscore the effectiveness of embedded time series and the importance of the TSFM as a beneficial temporal feature extractor. We additionally provide analysis on the scenarios where the TSFM is the most beneficial in Appendix D.1.

# 5 Conclusion

We introduce TS-Reasoner, a framework that advances the ability of LLMs to understand and reason about time series by bridging with TSFM. To mitigate the intrinsic semantic gap, we further developed an attribute-aware captioning method that enriches time-series alignment data, fostering a more robust alignment. Extensive experiments demonstrate that TS-Reasoner substantially outperforms a wide range of baselines on time series understanding and reasoning benchmarks.

**Discussion.** In practice, the choice between TS-Reasoner and proprietary models such as GPT-4.1 depends on deployment constraints. In domains such as finance and healthcare, data privacy regulations often prohibit sending time series to external cloud services, ruling out API-based proprietary models. As a fully open-source model that can be deployed locally, TS-Reasoner is well-suited to these settings, while requiring substantially lower computational cost than large proprietary multimodal models. This makes TS-Reasoner a practical choice when privacy, cost, or domain-specific adaptability are important considerations.

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

## A    TimesFM for time series embedding

Given a time series $\mathcal{T} \in \mathbb{R}^L$, where $L$ is the length of the time series, we first normalize it to have a mean of zero and a variance of one. We then segment $\mathcal{T}$ into consecutive, non-overlapping patches of fixed length $P$, resulting in a total of $N = \lfloor L/P \rfloor$ patches. This yields a patched time series $\mathcal{T}_p \in \mathbb{R}^{N \times P}$.

Following the approach of (Das et al., 2024), the $j$-th patch $\mathcal{T}_p^j$ is passed through a residual block to project it into the model dimension. This block is implemented as a two-layer MLP with a skip connection, processing each patch independently. The input token for the $j$-th patch is computed as:

$$\mathcal{E}_p^j = \text{InputResidualBlock}\left(\mathcal{T}_p^j\right) + \text{PE}_j, \tag{6}$$

where $\text{PE}_j$ is the position encoding for the $j$-th patch, as defined in the original transformer (Vaswani et al., 2017). These encoded patch representations are then fed into an $M$-layer stacked Transformer to produce the final sequence of time series features:

$$\mathcal{Z}_T = \text{StackedTransformer}([\mathcal{E}_p^{(0)}, \mathcal{E}_p^{(1)}, ..., \mathcal{E}_p^{(N)}]), \tag{7}$$

where $\mathcal{Z}_T \in \mathbb{R}^{N \times d_{\text{ts}}}$ and $d_{\text{ts}}$ denotes the embedding dimension for each time series patch. Refer to more details of TimesFM in (Das et al., 2024).

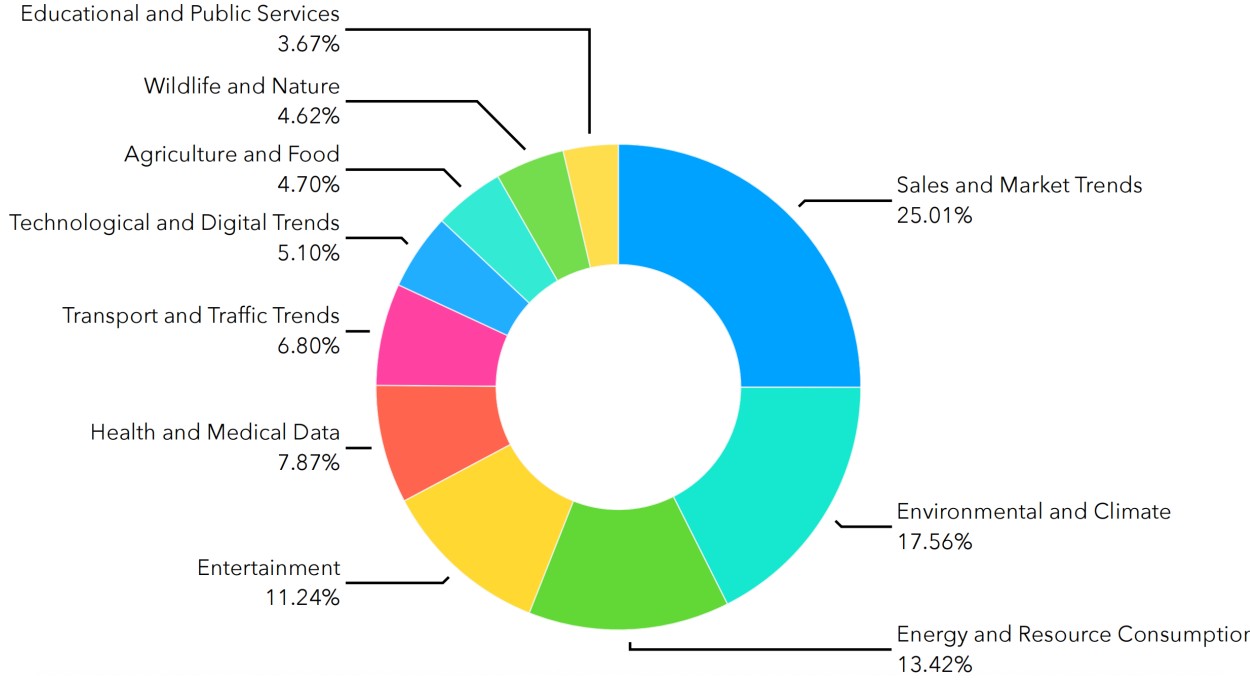

Figure 9: Domain distribution of LLM-generated time series with context.

## B    Qualitative Analysis of Captioning

To qualitatively evaluate the distinct advantages of our approach, we conduct a case study comparing three methods: (1) our proposed attribute-aware captioning, which leverages visual time series plots and explicit attribute guidance; (2) a basic captioning baseline that operates on visual plots but lacks attribute guidance; and (3) LLM prompted with the raw textual (numerical) time series data. Our analysis, illustrated in Figure 10, yields two key insights.

(i) **Attribute-Aware Captions Provide Semantically Richer Descriptions.** A primary limitation of basic captioning is its tendency to produce superficial, chronological narrations of the data. As shown

in Figure 10, the captioner describes the series' movements (e.g., "the value increases, then decreases sharply") but fails to extract deeper, underlying characteristics. While factually correct, this description omits properties crucial for a comprehensive understanding. In contrast, our attribute-aware captioning enriches this chronological account with critical semantic attributes. It not only captures the temporal dynamics but also identifies and articulates the series' overall trend, periodicity, and noise level. This multifaceted analysis provides a more holistic understanding of the time series, which is essential for TS-REASONER to conduct reasoning on downstream tasks.

(ii) **Visual Representation is Crucial for Capturing Global Temporal Patterns.** When comparing our visually-grounded method to an LLM processing raw numerical data, a significant gap emerges in the ability to identify global patterns. The text-based LLM, while capable of discerning local features like high-frequency oscillations or noise within a limited window, consistently fails to recognize the overarching periodicity of the entire series. We hypothesize that this failure stems from the inherent inefficiency of representing long numerical sequences as text. The excessive length of input may distract the LLM, preventing it from observing the complete pattern. Conversely, a time series plot serves as a highly compressed, holistic representation. It enables the model to perceive the entire sequence as a single input, making global structures like periodicity visually salient and readily identifiable.

## C Comparison with Related LLM–Time-Series Architectures

Recent work couples large language models (LLMs) with time series in several distinct ways. Table 8 contrasts representative methods along architecture, training data, which components are frozen versus fine-tuned, and the evaluation tasks they target.

These methods differ in both the direction of knowledge transfer and the granularity of the time-series representation handed to the LLM. ChronoSteer shows that LLM knowledge can be injected into a TSFM to strengthen forecasting, transferring information in the LLM to the TSFM direction. Conversely, TS-Reasoner demonstrates the complementary direction: Integrating TSFM knowledge into the LLM improves the LLM's ability to understand time series. The two remaining baselines differ from ours in representation granularity. TempoGPT tokenizes time series into discrete tokens via a quantization codebook, whereas TS-Reasoner aligns the continuous embeddings of a pretrained TSFM with the LLM directly, avoiding the information loss induced by quantization. Time-VLM renders the time series into images and pair them with textual prompts to exploit a frozen VLM's vision-language alignment for forecasting. In contrast, TS-Reasoner grounds the LLM in the continuous representations of a pretrained TSFM, showing that this is an effective route to time-series understanding without relying on visualization.

## D Additional Experimental Results

### D.1 Scenarios Where TSFM is Most Beneficial

The ablation in Table 6 shows that removing the TSFM consistently degrades performance, though by a smaller margin than removing instruction tuning. To clarify when the TSFM contributes most, we analyze its effect along two axes.

*(1) By input length.* Table 9 reports the per-subset accuracy drop on MTBench together with the average input length of each subset.[1] Within each domain, the drop grows with input length: in finance, removing the TSFM costs 2.15 points at an average length of 375 steps versus 0.69 at 135; in weather, 1.79 at 336 versus 0.62 at 168. The TSFM is thus increasingly beneficial for longer series, where its patch-level representations compactly summarize long-range temporal structure that the LLM struggles to recover from raw value sequences.

*(2) By modality reliance.* Removing the TSFM reduces overall accuracy by 3.07% on TimeSeriesExam but only 2.22% on average on MTBench (Table 6). We attribute this gap to how strongly each benchmark relies

---

[1]In MTBench, the short/long split refers to the question horizon (7-day vs. 14-day prediction), not the input length; input lengths vary independently across subsets.

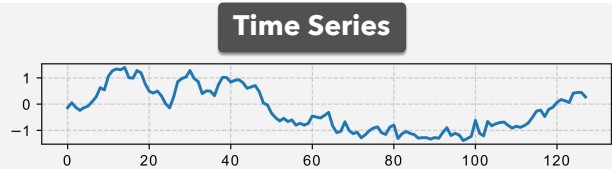

**Input:** Create a detailed description of the time series in one paragraph.

**Output:** The time series plot shows a fluctuating trend with distinct phases of increase and decrease over time. Initially, the series begins near zero and exhibits a sharp upward movement, peaking around the 10–20 time mark, where it reaches values greater than 1. This is followed by a volatile but generally declining trend that persists through to approximately the 60th time unit, dipping below -1. From this low point, the series remains subdued and oscillates around a low mean until about time step 100. After that, a gradual upward recovery begins, with the series rising back toward positive values and ending slightly above zero near time step 130. The fluctuations suggest a non-stationary process with alternating periods of high volatility and relative stability.

**Input with visualized time series:** Create a detailed description of the time series in one paragraph, including its trend, frequency properties, periodicity, noise, local variations, and other relevant characteristics.

**Output:** This time series displays a **prominent, low-frequency pattern without a clear long-term linear trend.** The series appears **non-stationary**, as its mean value shifts significantly over time, completing roughly one full oscillation over the observed interval of approximately 125 time units. The primary cycle involves a rise to a peak around time unit 20, followed by a decline to a trough near time unit 85, and a subsequent recovery. Superimposed on this dominant wave are higher-frequency, smaller-amplitude fluctuations, which introduce **considerable noise** and give the line a jagged, irregular texture. The volatility seems relatively constant, although there are **local variations in the magnitude and sharpness of these smaller oscillations**. Strong positive autocorrelation is evident, as consecutive data points are closely related, resulting in a continuous, flowing path rather than random jumps.

**Input with textual time series:** Here is a time series: [0.13, 0.05, -0.12, -0.24, ...]. Create a detailed description of the time series in one paragraph, including its trend, frequency properties, periodicity, noise, local variations, and other relevant characteristics.

**Output:** The time series exhibits a strong quasi-periodic behavior without a discernible long-term trend, characterized by a distinct cycle of a sharp rise to a peak, a steep fall into a trough, and a subsequent gradual recovery. This dominant low-frequency pattern is superimposed with considerable high-frequency noise and local variations, resulting in fluctuations that make each cycle similar but not identical. Overall, the data represents a noisy cyclical signal, where a clear underlying periodicity is modulated by random, short-term movements. ✗

Figure 10: Case study comparing time series captioning methods. Basic visual captioning (row 1) gives a chronological description. Our attribute-aware visual captioning (row 2) provides a richer description including key attributes. Textual time series captioning (row 3) identifies some attributes but fails to capture the periodicity pattern.

on the time-series modality: TimeSeriesExam questions depend almost entirely on the time series itself, so weaker temporal representations are directly penalized, whereas MTBench questions can additionally draw on contextual news or weather reports that partially compensate. The TSFM therefore contributes most when reasoning hinges primarily on the time series, and more modestly when complementary textual context is available.

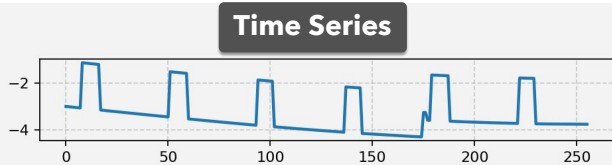

**Input:** Please analyze the characteristics of this time series.

**Template Caption:** The length of the time series is 256. From the perspective of the slope, the overall trend is decreasing. However, local phase changes were observed, including: sudden increase. The value of time series starts from around -3.00 and ends at around -3.76, with an overall amplitude of -0.76. The amplitude of the periodic fluctuation is approximately 1.9, showing periodic fluctuation. Each fluctuation period is approximately 42.9 points, thus the overall fluctuation is low frequency. The overall noise standard deviation is around 0.00, very small compared the overall change of the curve. The curve is overall smooth with almost no noise. In terms of local characteristics, a sudden increase with an amplitude of 1.05 occurred between point 174 and point 176, with the time series value rising from around -4.29 to around -3.26, then a drop with an amplitude of 0.32 occurred between point 176 and point 177, with the time series value falling back to around -3.59, forming a sudden increase.

**Input:** Create a detailed description of the time series in one paragraph, including its trend, frequency properties, periodicity, noise, local variations, and other relevant characteristics.

**LLM-generated Caption:** This series is a clean, deterministic pulse train with a strongly asymmetric cycle: each period begins with a near-instantaneous step up to a short, flat high plateau (about 8–9 steps wide, a duty cycle of roughly 20%), followed by an equally abrupt drop and then a long, smooth, gently downward-sloping decay ramp that runs until the next pulse fires. The pattern is quasi-periodic, with a fundamental period of about 43 steps, placing the dominant spectral component at f ≈ 0.023 cycles/step. Because the pulse edges are nearly vertical and the inter-pulse ramps are sawtooth-like, the spectrum is not a single tone but a low fundamental trailed by a long tail of slowly decaying harmonics—the broadband high-frequency energy typical of square/sawtooth waveforms—and the autocorrelation would combine a slow, ramp-driven decay with secondary peaks at lags that are multiples of ~43. Stochastic noise is effectively absent: the plateaus and ramps are smooth, and the only irregularity is a small glitch-like double feature around t ≈ 174–178. Two slow non-stationary trends ride on top of the periodicity: the plateau heights fall monotonically over the first four cycles and then partially recover, while the baseline drifts down from ≈ −3.0 to a minimum of ≈ −4.3 near t ≈ 174 before reverting toward ≈ −3.7, giving the overall level a shallow U-shaped envelope rather than a monotone drift. In short, it is a low-noise, asymmetric pulse/sawtooth process with a ~43-step period, modulated by a slowly varying, mean-reverting amplitude and baseline.

Figure 11: Case study comparing time series captioning methods. Template-based captioning (row 1) fills a fixed set of predefined attributes with rigid, repetitive phrasing. Our LLM-generated captioning (row 2) offers greater language diversity and a more comprehensive, flexible description that is not restricted to predefined attributes.

## E  Prompts

Below, we detail all the prompts we used. Figure 12 shows our captioning prompts. Figure 13 shows the prompt for the TimeSeriesExam benchmark. Figure 14 and Figure 15 show the prompt for MTBench.

## F  Data Leakage Examination

To guarantee evaluation integrity and prevent data leakage, particularly given LLM-synthesized data, we strictly prevent data leakage at both the source and semantic levels. We enforce source isolation by deriving training data exclusively from synthetic sources (Ansari et al., 2024) and public archives (Merrill et al., 2024) for alignment, and the instruction tuning dataset in (Xie et al., 2024), explicitly verifying no numerical

**Instructions for prompting LLMs to generate time series captions.**

1. Write a paragraph that analyzes the time series, covering its local behaviors, noise levels, periodic structures, overall trend, frequency content, and any other characteristics you consider important.

2. Create a detailed description of the time series in one paragraph, including its trend, frequency properties, periodicity, noise, local variations, and other relevant characteristics.

3. Provide a paragraph summarizing the time series characteristics such as noise, periodic patterns, long-term trends, frequency behavior, local anomalies, and any other significant features.

4. Compose a detailed caption describing the frequency characteristics, noise, trends, local variations, periodic structures, and any other meaningful patterns you observe in the time series.

5. Craft a one-paragraph summary of the time series, noting local fluctuations, periodic behavior, frequency features, trend, noise content, and any other insights you find important.

6. Generate a descriptive paragraph detailing the time series' key attributes, including frequency structure, noise patterns, trend direction, local features, periodic elements, and other notable aspects.

7. Give a thorough one-paragraph explanation of the time series, addressing periodicity, noise, frequency components, trend, local variations, and other relevant characteristics.

8. Write a narrative paragraph explaining the time series, focusing on noise, frequency characteristics, periodicity, localized structures, the overall trend, and other important features you identify.

9. Summarize the time series in a paragraph, describing its fluctuations, recurring patterns, noise levels, frequency-domain features, trend direction, and any additional traits you find significant.,

10. Develop a paragraph that captures the key features of the time series, such as frequency traits, trend, noise, periodic components, local behaviors, and other characteristics worth noting.

11. Provide a one-paragraph caption analyzing the time series data in terms of noise, trend, periodicity, local features, frequency-related behavior, and any additional characteristics of interest.

12. Create a rich paragraph description of the time series, including its trend, local anomalies, periodic activity, noise artifacts, spectral content, and other important descriptive elements.

13. Write a descriptive paragraph for the time series, highlighting frequency properties, trend behavior, periodic patterns, local structures, noise, and other characteristics you consider relevant.

14. Generate a compact yet thorough paragraph explaining the time series in terms of periodicity, trend movement, noise level, frequency details, local dynamics, and any other key aspects.

15. Construct a one-paragraph analysis of the time series by examining its local variations, noise, trend, periodic elements, frequency spectrum, and other notable features you deem important.

16. Write a summary paragraph that discusses the time series' periodic features, trend behavior, local patterns, noise levels, frequency domain signals, and other characteristics worth mentioning.,

17. Create a detailed one-paragraph commentary on the time series that outlines its noise characteristics, periodicity, frequency content, trends, localized behaviors, and other useful insights.

18. Prepare a paragraph-long description of the time series covering its trend, noise, frequency-related traits, local fluctuations, periodic structures, and any additional attributes of note.,

19. Offer a one-paragraph interpretation of the time series, highlighting its frequency features, periodic nature, local patterns, noise, trend line, and any other important characteristics you observe.

20. Compose a detailed summary in one paragraph focusing on the time series' periodic behavior, frequency spectrum, localized fluctuations, overall trend, noise, and other relevant descriptive elements.

Figure 12: The list of instructions for attributes-aware time series captioning.

Table 7: Example template questions for different reasoning tasks. Each subcategory covers a specific aspect of time series understanding, guiding the model to reason about comparative, anomalies, and causal relationships.

| Category | Subcategory | Example question |
|---|---|---|
| Pattern Recognition | Trend | What is the most likely linear trend coefficient of the given time series? |
| | Cyclic | The given time series has a sine wave pattern. How does its amplitude change from the beginning to the end? |
| | Stationarity | Is the given time series likely to be stationary after removing the cycle component? |
| | Regime Switching | Based on the given time series, how many different regimes are there? |
| | Statistical properties | Is the mean stable over time in the given time series? |
| | Random processes | Does the following time series exhibit a mean reversion property? |
| Noise Understanding | White Noise | Is the given time series a white noise process? |
| | Random Walk | Is the given time series likely to be a random walk process? |
| | Signal / Noise Ratio | You are given two time series with the same underlying pattern but different noise levels. Which time series has a higher magnitude of noise? |
| Anomaly Detection | | The following time series has two types of anomalies appearing at different time points. What are the likely types of these anomalies? |
| Similarity Analysis | Shape | Despite the noise, do the two given time series have similar patterns? |
| | Distributional | You are given two time series, which are generated using a random walk. Are they likely to have the same variance? |
| Causality Analysis | Granger Causality | Is there Granger causality between the two time series? |

> **Instructions for prompting LLMs for time series understanding and reasoning tasks in Timeseriesexam.**
>
> You are a time series analysis expert. These are the time series data: $\langle ts_1 \rangle \langle ts_1/ \rangle \ldots \langle ts_k \rangle \langle ts_k/ \rangle$. Please answer the question and provide the correct option letter.
> Question: <Question>
> Choices: <choices>

Figure 13: Prompt example for time series understanding and reasoning in Timeseriesexam.

overlap with the MTBench and TimeSeriesExam benchmarks. Furthermore, we ensure narrative and template isolation; our synthesized training captions utilize attribute-aware prompts focusing on generic properties like trends and periodicity, which are structurally and semantically distinct from the domain-specific

| Method | Architecture | Training Data | Frozen / Fine-tuned | Evaluation Tasks |
|---|---|---|---|---|
| ChronoSteer (Wang et al., 2025a) | LLM emits textual revision instructions to steer a TSFM's forecast (LLM→TSFM) | Synthetic instruction–series data | LLM and TSFM frozen; alignment module trained | Time series forecasting |
| TempoGPT (Zhang et al., 2025a) | VQ-VAE quantizes the series into tokens; the LLM vocabulary is expanded with a shared embedding layer | Electrical time series simulation dataset | VQ-VAE frozen; LLM fine-tuned | Self-built electrical time series reasoning tasks |
| Time-VLM (Siru et al., 2025) | Time series encoded as images via a pretrained VLM, then fused for forecasting | Public time series forecasting datasets | VLM frozen; fusion network trained | Time series forecasting |
| **TS-Reasoner (ours)** | Frozen pretrained TSFM features projected into the LLM via an adapter for reasoning (TSFM→LLM) | Cross-domain captions + instructions | TSFM frozen; adapter + LLM fine-tuned | Public reasoning benchmarks + open-ended tasks |

Table 8: Comparison of representative LLM–time-series architectures. Unlike methods that steer a forecaster with text or quantize series into discrete tokens, TS-Reasoner aligns the *continuous* embeddings of a frozen TSFM with the LLM through a lightweight adapter, targeting reasoning rather than forecasting alone.

Table 9: Effect of removing the TSFM on MTBench, broken down by domain and average input length (number of time steps). $\Delta_{\text{TSFM}}$ denotes the accuracy change of the "– TSFM" ablation relative to the full TS-Reasoner (cf. Table 6). Within each domain, subsets with longer input series incur larger drops. The "short"/"long" labels denote the temporal horizon of the questions (7-day vs. 14-day).

| Domain | Subset | Avg. time series length | $\Delta_{\text{TSFM}}$ |
|---|---|---|---|
| Finance | short | 375 | −3.58 |
|  | long | 135 | −2.57 |
| Weather | long | 336 | −1.79 |
|  | short | 168 | −0.93 |

financial analyses and weather reports in MTBench and the systematic property evaluation questions in TimeSeriesExam.

In addition, we evaluate the potential overlap between the instruction-tuning data and the evaluation benchmarks by following (Wei et al., 2021; Brown et al., 2020; Du et al., 2022), where an example from the evaluation benchmark is considered contaminated if any 13-gram in it also appears in the instruction-tuning data. Results in Table 10 show that no example contains a 13-gram that appears in our Stage-2 instruction-tuning corpus. This 0% contamination rate provides evidence that Stage 2 drives genuine reasoning capabilities rather than the memorization of benchmark-specific formats.

Table 10: Contamination ratio between instruction-tuning data and evaluation benchmark.

| Benchmark | TimeSeriesExam | MTBench QA-long (finance) | MTBench QA-long (weather) | MTBench QA-short (finance) | MTBench QA-short (weather) |
|---|---|---|---|---|---|
| **Rate** | 0.00% | 0.00% | 0.00% | 0.00% | 0.00% |

---

**Instructions for prompting TS-REASONER for financial reasoning tasks in MTBench.**

You are an expert in finance and stock market analysis. Your task is to answer the question based on the given n-day historical stock price time series and a financial analysis published at the last timestamp of the time series. The time series is: $\langle ts_1 \rangle \langle ts_1/ \rangle$.
<Context>
Question: <Question>
Choices: <Choices>

Figure 14: Prompt example for time series reasoning in Finance.

---

**Instructions for prompting TS-REASONER for weather reasoning tasks in MTBench.**

You have an n-day temperature time series, and a weather event report published on the last day of the time series. The time series below is the 14-day temperature between <start time> to <end time>, and the time interval is 1 hour: $\langle ts_1 \rangle \langle ts_1/ \rangle$.
The following events are reported:
<Context>
Question: <Question>
Choices: <Choices>

Figure 15: The prompt for time series reasoning in Weather.

