# OpenReview forum: "TS-Reasoner: Aligning Time Series Foundation Models with LLM Reasoning"
_TMLR — Under review for TMLR_

### Review · Reviewer_8ZbR · 2026-04-29

**Summary Of Contributions:**

This paper proposes TS-Reasoner, a time-series large language model that aligns a pretrained Time Series Foundation Model with an LLM for time series understanding and reasoning. Instead of converting time series into raw numerical strings or relying solely on visual plots, TS-Reasoner encodes time series with a frozen TSFM and projects the resulting temporal representations into the LLM embedding space through a trainable TS-to-Text adapter. A key contribution is a two-stage training recipe: first, alignment pretraining with time-series captioning data, and then instruction tuning for downstream reasoning tasks. To address the scarcity of time-series–text pairs, the paper introduces an attribute-aware caption synthesis method, using advanced LLM/VLMs to generate diverse captions that describe trends, periodicity, noise, anomalies, and other temporal properties. Empirically, TS-Reasoner outperforms comparable open-source LLMs, VLMs, and time-series LLM baselines on time series understanding and reasoning benchmarks, while showing improved data efficiency. Ablation studies further support the importance of the pretrained TSFM, caption-based alignment, attribute-aware caption generation, and the two-stage training pipeline.

**Audience:**

Yes

**Audience Explanation:**

The findings should be of interest to at least part of the TMLR audience, especially researchers working on multimodal learning, time series foundation models, LLM-based reasoning, and synthetic data for model alignment. The paper demonstrates a practical way to connect pretrained TSFMs with LLMs and provides useful empirical evidence on caption-based alignment for time series.

**Claims And Evidence:**

Yes

**Claims Explanation:**

The claims are generally supported by clear empirical evidence, especially through comprehensive comparisons and ablation studies. The results convincingly show that the proposed combination of TSFM-based encoding, caption-based alignment, and instruction tuning improves performance over comparable open-source LLM/VLM/TSLLM baselines.

**Requested Changes:**

1. The authors should discuss potential data overlap or distributional similarity between the instruction-tuning data and the evaluation benchmarks. Since Stage 2 instruction tuning contributes substantially to performance, it is important to clarify whether the gains reflect general time-series reasoning ability or adaptation to benchmark-specific question formats.

2. The role of the TSFM should be examined more deeply. The ablation shows that removing the TSFM reduces performance, but the margin is moderate compared with the effect of instruction tuning. Additional analysis would help clarify when the TSFM is most beneficial, such as by breaking down results by sequence length, task type, domain, or cases requiring fine-grained temporal pattern recognition.

3. The authors should address the implications of instance normalization. While normalization improves robustness to shifts and scales, it may remove absolute magnitude information that is important for threshold-based reasoning, such as temperatures below freezing, financial price levels, or abnormal values relative to fixed domain constraints. A discussion or ablation on normalization would make the method more transparent.

---

> ### Author Response · Authors · 2026-06-14
> **Response to Reviewer 8ZbR**
>
> We thank the reviewer 8ZbR for the detailed and constructive feedback. We address each point in turn below, and indicate the corresponding revisions we will make to the manuscript.
>
> __Q1__: Discussion on potential data overlap or distributional similarity between the instruction-tuning data and the evaluation benchmarks.
>
> __A1__: Thank you for your suggestion. We follow your suggestion to evaluate the potential data overlap between the instruction-tuning data and the evaluation benchmarks. Specifically, we follow the settings in [1, 2, 3], where an example from the evaluation benchmark is considered contaminated if any 13-gram in it also appears in the instruction-tuning data. We report the ratio of overlap data in the evaluation dataset:
> | Benchmark | Rate |
> | :--- |  :--- |
> | TimeSeriesExam | 0.00% |
> | MTBench QA-long (finance) | 0.00% |
> | MTBench QA-long (weather) |  0.00% |
> | MTBench QA-short (finance) |0.00% |
> | MTBench QA-short (weather) | 0.00% |
>
> Across all five benchmarks, no example contains a 13-gram that appears in our Stage-2 instruction-tuning corpus. This 0% contamination rate provides evidence that Stage 2 drives genuine reasoning capabilities rather than the memorization of benchmark-specific formats. These results are incorporated into Appendix F.
>
> __Q2__: The ablation shows that removing the TSFM reduces performance, but the margin is moderate compared with the effect of instruction tuning. Additional analysis would help clarify when the TSFM is most beneficial, such as by breaking down results by sequence length, task type, domain, or cases requiring fine-grained temporal pattern recognition.
>
> __A2__: Thank you for your insightful comment. In TS-Reasoner, TSFM plays a role in providing richer time series features for the backbone LLM, and instruction tuning enables the model to leverage such information for reasoning.
>
> We further analyze the benefits brought by TSFM:
>
> (1) By sequence length: we analyze the TSFM's effect with respect to time series length on MTBench. As shown below, the performance drop from removing the TSFM broadly increases with the sequence length, suggesting that the TSFM is increasingly beneficial as time series grow longer:
>
> | MTBench | Average time series length | TSFM Δ |
> |---|---|---|
> | Finance (short) | 375 | −2.15 |
> | Weather (long) | 336 | −1.79 |
> | Weather (short) | 168 | −0.62 |
> | Finance (long) | 135 | −0.69 |
>
> (2) By task type and domain: As shown in Table 6, removing the TSFM leads to a 2.5% overall drop on TimeSeriesExam but only a 1.3% drop on MTBench. We attribute this to their different reliance on time series. TimeSeriesExam questions depend almost entirely on the time series itself, so weaker temporal representations lead to larger performance drops. MTBench questions can also draw on contextual news/weather reports, which partially compensate for weaker temporal representations. This indicates that the TSFM is most beneficial when reasoning relies primarily on the time series, and its contribution becomes more modest when complementary textual context is available.
>
> We incorporate the above discussion and results into our Appendix D.1.
>
> __Q3__: While normalization improves robustness to shifts and scales, it may remove absolute magnitude information that is important for threshold-based reasoning, such as temperatures below freezing, financial price levels, or abnormal values relative to fixed domain constraints. A discussion or ablation on normalization would make the method more transparent.
>
> __A3__: Thank you for your insightful question.
> To preserve absolute magnitude information, we adopt a value-preserved time-series normalization scheme. Specifically, for each input series $x \in \mathbb{R}^L$, we compute the mean $\mu = \frac{1}{L}\sum_{t=1}^{L} x_t$ and subtract it: $\tilde{x} = x - \mu$. When the centered signal's magnitude exceeds a fixed stability range, we further divide it by a per-series scale factor $s$ to obtain the normalized series $\hat{x} = \tilde{x} / s$.
>
> The resulting offset value and scaling value are then prepended to the time-series placeholder as plain-text numerical tokens, followed by the <ts><ts/> tokens. The LLM therefore sees both the normalized patch features encoded by the TSFM and the raw normalization parameters, from which the original magnitude of any value can be exactly recovered. This approach enables the model to perform threshold-based reasoning (e.g., sub-freezing temperatures). We made make this design explicit in §3.3 of the revised paper.
>
>
>
> References:
>
> [1] Wei, Jason, et al. "Finetuned Language Models are Zero-Shot Learners." International Conference on Learning Representations.
>
> [2] Brown, Tom, et al. "Language models are few-shot learners." Advances in neural information processing systems 33 (2020): 1877-1901.
>
> [3] Du, Nan, et al. "Glam: Efficient scaling of language models with mixture-of-experts." International conference on machine learning. PMLR, 2022.

---

### Review · Reviewer_FzQt · 2026-05-14

**Summary Of Contributions:**

This paper proposes TS-Reasoner, incorporating latent representations of time series data together with textual inputs for improving the performance on understanding/reasoning tasks on time series data. Specifically, this paper develops a two stage training approach
1) Pre-training for Language-Timeseries Alignment: given the time series data (represented as latent representation), output a text summary. For this stage, the (time series, text summary) data pairs are produced by a teacher model, e.g. GPT-4.1.
2) Instruction Finetuning for Time Series Reasoning: standard SFT on specific time series Q/A or reasoning tasks.

This paper studies an interesting and important area: how to better understand time series data for LLMs. And experiments show that the proposed approach surpasses both the base LLM and other time series LLMs.

**Audience:**

Yes

**Audience Explanation:**

The paper addresses an important problem of better understanding time series data for LLM. This is traditionally a challenge because LLMs are not super good at numbers. So this will be a topic many will be interested in.

**Claims And Evidence:**

Yes

**Claims Explanation:**

The biggest concern I have is unfair comparison.

For comparing with the base LLM, the two stage training approach is effectively doing 1) some form of distillation from a better teacher model GPT-4.1 in the first stage, 2) domain specific SFT in the second stage. So it won't be a surprise the final performance is better than the base LLM Qwen2.5-7B-Instruct. For a comparison to demonstrate the effectiveness of the time series component, a more meaningful baseline will be doing the same two stage finetuning on the base model: without the time series component, just pass the time series as text (or image plot). This should help answer whether the time series component is indeed helpful, or the metric gains come from the additional data (distillation data from teacher + domain SFT data)

For comparing with other time series LLM, a similar question is that the data used are different since this paper uses additional distillation caption data from a better teacher model.

**Requested Changes:**

1. The numbers don't match in table 2 and table 3. "TS-Reasoner-7B (ours)" row in table 2 should match the "TimesFM" row in table 3, right? Actually, table 2, figure 6, figure 7 seem to use one set of numbers but table 3, figure 8, table 6 seem to use another set of numbers.

2. For "∆ Over Best 7B" in table 2, NU reports +8.28. 60.15 - 8.28 = 51.87 but I don't see anything close to 51.87 for other 7B models.

3. "See more implementation details in Appendix 4." in section 4 but there's no Appendix 4 (only Appendix ABCD) and I don't see those implementation details anywhere.

4. In abstract, it says "TS-Reasoner not only outperforms a wide range of prevailing LLMs". It should be more clearly stated the metric gains are over similar size open source models since experiments clearly show the performance is worse than models like GPT-4.1 and DeepSeek-R1.

5. Wrong reference to "Ministral-8B-Instruct". Current reference points to an older Mixtral paper.

---

> ### Author Response · Authors · 2026-06-14
> **Response to Reviewer FzQt**
>
> We thank the reviewer FzQt for the constructive and thoughtful feedback. Below, we provide detailed responses to each of your comments and hope to address any further considerations you may have.
>
> __Q1__: Adding comparison with the base LLM which trains with the same process as TS-Reasoner. For comparing with other time series LLM, a similar question is that the data used are different since this paper uses additional distillation caption data from a better teacher model.
>
> __A1__: Thank you for your insightful question.
>
> (i) *Regarding the comparison with base LLM trained with the same process as TS-Reasoner*: We follow your suggestion to train the Qwen2.5-7B-Instruct using the same training pipeline as TS-Reasoner, which inputs time series as raw text. The results are shown as follows:
>
> | Model | PR | NU | AD | SA | CA | OA | Finance (long) | Finance (short) | Weather (long) | Weather (short) |
> | :--- | :---: | :---: | :---: | :---: | :---: | :---: | :---: | :---: | :---: | :---: |
> | Qwen2.5-7B-Instruct  | 47.17 | 47.13 | 41.86 | 53.10 | 41.27 | 46.66 | 87.98 | 89.41 | 57.14 | 58.44 |
> | Qwen2.5-7B-Instruct (trained with TS-Reasoner pipeline) |48.25 | 55.17 | 42.64 | 53.98 | 50.45 | 49.67 | 88.95 | 90.43 | 58.10 | 59.38 |
> | TS-Reasoner | 53.46 |60.15 | 53.23 | 63.42 | 43.39 | 54.83 | 92.00 | 93.28 | 60.44 | 61.55 |
>
> The base LLM trained with the identical pipeline has performance gain, but still underperforms TS-Reasoner across all benchmarks. This confirms that the improvements stem from our time series components rather than the training data alone.
>
> (ii) *Regarding the comparison with other time series LLMs trained with the same data as TS-Reasoner*: We follow your suggestion to train two other time series LLMs, including ChatTime-7B and ChatTS-7B. The results are shown as follows:
>
> | Model | PR | NU | AD | SA | CA | OA | Finance (long) | Finance (short) | Weather (long) | Weather (short) |
> | :--- | :---: | :---: | :---: | :---: | :---: | :---: | :---: | :---: | :---: | :---: |
> | ChatTime-7B | 45.55 | 51.72 | 41.08 | 48.67 | 36.51 | 45.21 | 34.88 | 32.18 | 50.54 |48.43 |
> | ChatTS-7B |50.94 | 55.17 | 51.94 | 55.75| 34.92 | 50.98 | 88.57 | 89.21 | 58.10 | 59.69 |
> | TS-Reasoner | 53.46 | 60.15 | 53.23 | 63.42 | 43.39 | 54.83 | 92.00 | 93.28 | 60.44 | 61.55 |
>
> The results show that TS-Reasoner still outperforms other time series LLMs when trained on the same training data.
>
> __Q2__: "TS-Reasoner-7B (ours)" row in table 2 should match the "TimesFM" row in table 3, right? Actually, table 2, figure 6, figure 7 seem to use one set of numbers but table 3, figure 8, table 6 seem to use another set of numbers.
>
> __A2__: Thank you for your question. Yes, the "TS-Reasoner-7B (ours)" row in Table 2 and the "TimesFM" row in Table 3 correspond to the same model configuration, and the numerical discrepancy comes from different reporting protocols: Table 2, Figure 6, and Figure 7 report the accuracy averaged over three independent inference runs, whereas Table 3, Figure 8, and Table 6 report the result of a single run. We unify all results to the three-run average in the revision for consistency.
>
> __Q3__: For "∆ Over Best 7B" in table 2, NU reports +8.28. 60.15 - 8.28 = 51.87 but I don't see anything close to 51.87 for other 7B models.
>
> __A3__: Thank you for pointing out this typo. The "∆ Over Best 7B" row is computed against the best-performing baseline of comparable size, which for NU is InternVL3-8B with 52.87. The correct value is therefore 60.15 − 52.87 = +7.28 rather than +8.28. We correct this typo in the revision.
>
> __Q4__: "See more implementation details in Appendix 4." in section 4 but there's no Appendix 4 (only Appendix ABCD) and I don't see those implementation details anywhere.
>
> __A4__: Thank you for your comments. "Appendix 4" was intended to refer to the "Implementation Details" paragraph and Table 1 in Section 4, where the implementation details are provided. We will remove this sentence in the revision.
>
> __Q5__: In abstract, it says "TS-Reasoner not only outperforms a wide range of prevailing LLMs". It should be more clearly stated the metric gains are over similar size open source models since experiments clearly show the performance is worse than models like GPT-4.1 and DeepSeek-R1.
>
> __A5__: Thank you for your suggestion. We will revise the sentence in the abstract to: *Extensive experiments on several benchmarks demonstrate that TS-Reasoner not only outperforms a wide range of open-source LLMs, Vision Language Models (VLMs), and Time Series LLMs of comparable scale, but also achieves this with remarkable data efficiency, e.g., using less than half the training data*. We revise it in the abstract.
>
> __Q6__: Wrong reference to "Ministral-8B-Instruct". Current reference points to an older Mixtral paper.
>
> __A6__: Thank you for your comments. We replace it with the correct reference to Ministral-8B-Instruct in the revision.

---

> > ### Comment · Reviewer_FzQt · 2026-06-18
> >
> > Thanks for the response. Most of my concerns have been addressed and I will update my ratings to Yes and Yes.

---

### Review · Reviewer_BWyT · 2026-06-06

**Summary Of Contributions:**

This paper introduces TS-Reasoner, a framework that combines Time Series Foundation Models (TSFMs) and Large Language Models (LLMs) to improve time series reasoning. This is a task that requires both numerical temporal understanding and contextual/linguistic reasoning. The core idea is to use a pretrained TSFM as a frozen feature extractor that encodes time series data into rich temporal embeddings, which are then projected into an LLM's input space via a single layer MLP adapter, allowing the LLM to reason over both numerical patterns and textual context simultaneously. To address the scarcity of training data, the authors develop an "attribute-aware captioning" method that uses advanced VLMs/LLMs (e.g., GPT-4.1) to generate diverse, high-quality textual descriptions of time series visualizations, and train the model in two stages: first an alignment pretraining stage using these synthetic captions, then an instruction fine-tuning stage for complex reasoning. Evaluated on the TimeSeriesExam and MTBench benchmarks, TS-Reasoner-7B consistently outperforms a wide range of baselines — including proprietary LLMs, open-source VLMs, and other time series LLMs — while achieving strong data efficiency, needing less than half the training data of comparable models to reach superior performance.

**Audience:**

Yes

**Audience Explanation:**

The framework presented by the paper yields good results, and is of interest to the community.

**Broader Impact Concerns:**

No impact concerns.

**Claims And Evidence:**

Yes

**Claims Explanation:**

The claims are supported by accurate, convincing and clear evidence. The paper includes a comparison with earlier methods.

**Requested Changes:**

- The “Attribute-aware captioning” paragraph is poorly written and confusing. In particular, how is the fundamental captioning instruction \mathcal{I}_{base} defined? Also, it is understood that Figure 11 shows the set of prompts, each of which should be a paraphrase of the augmented instruction, which contains attributes particular to the specific time series. However, these prompts are generic, short, and do not mention attributes specific to the time series. The whole paragraph has to be rewritten to make it clearer.

- It would be good to report the activation function and the width of the MLP adapter.

---

> ### Author Response · Authors · 2026-06-14
> **Response to Reviewer BWyT**
>
> We thank Reviewer BWyT for the helpful comments. We address each concern as follows:
>
> __Q1__: How is the fundamental captioning instruction $\mathcal{I}_{base}$ defined? Also, it is understood that Figure 11 shows the set of prompts, each of which should be a paraphrase of the augmented instruction, which contains attributes particular to the specific time series. However, these prompts are generic, short, and do not mention attributes specific to the time series. The whole paragraph has to be rewritten to make it clearer.
>
> __A1__: Thank you for your comments. The fundamental captioning instruction $\mathcal{I}_{base}$ is a manually written generic instruction, namely "*Create a detailed description of the time series in one paragraph.*" (shown in Figure 4). We would like to clarify that the attributes $\{a_1, \dots, a_G\}$ are a fixed set of $G=5$ generic attribute categories, including trend, frequency, periodicity, noise, and local variations, rather than instance-specific properties of each individual time series. These attributes are appended to $\mathcal{I}_{base}$ so that the captioning model knows which dimensions to analyze. The instance-specific attribute values are instead extracted by the captioning LLM itself from the visualized time series and written into the generated caption. We add more details in Section 3.4 accordingly.
>
> __Q2__: It would be good to report the activation function and the width of the MLP adapter.
>
> __A2__: Thank you for your suggestion. The TS-to-Text adapter is a single-layer MLP with a GELU activation, which projects the 1280-dimensional TSFM output features into the 3584-dimensional hidden space of the LLM. These details are added to the Implementation Details paragraph in Section 4.

---

### Review · Reviewer_fppB · 2026-06-07

**Summary Of Contributions:**

**Summary**

This paper addresses the problem of enabling LLMs to reason about raw time series data by aligning a frozen pretrained time series foundation model with an LLM. The authors propose TS-Reasoner, combining a frozen TimesFM-200M encoder with Qwen2.5-7B-Instruct via a single-layer MLP adapter, trained through a two-stage recipe: (1) alignment pretraining with 120K synthetic caption pairs (template-based and attribute-aware LLM-generated), and (2) instruction finetuning with 30K samples. TS-Reasoner-7B achieves 54.83% overall accuracy on TimeSeriesExam (vs. ChatTS-7B at 50.72%) and strong MTBench scores, with competitive results against GPT-4.1 on weather reasoning. The contribution is primarily algorithmic and empirical - proposing a VLM-inspired architecture for time series reasoning with a synthetic data pipeline.

**Strengths**

- The problem formulation is clean and well-motivated. Aligning pretrained temporal representations with language models through the VLM paradigm (frozen encoder + adapter + LLM) is conceptually accessible, and the paper communicates it effectively.

- The experimental evaluation includes a comprehensive baseline set spanning LLMs, VLMs, and time-series-specific LLMs, giving useful context for the contribution's positioning.

- The ablation study covers multiple dimensions - TSFM backbone choice (TimesFM vs. Chronos vs. MOMENT), training stage contributions, LLM scaling (0.5B/3B/7B), and captioning model quality - helping isolate what matters in the pipeline.

- The attribute-aware captioning approach with diversity metrics (MTLD scores, Self-BLEU-4) represents a genuine methodological contribution to synthetic data generation for time series alignment.

- The data efficiency story is compelling: TS-Reasoner outperforms ChatTS-7B with less than half the alignment data (120K vs. ~250K+).

- The inclusion of open-ended reasoning evaluation (Section 4.7, inductive reasoning task with RAGAS) goes beyond pure MCQA and shows consistent advantages over baselines including GPT-4o.

- The paper acknowledges its own limitations honestly, including MCQA evaluation constraints, template overfitting risks, and the modest TSFM ablation gap.

**Weaknesses**

1. The TSFM ablation shows only a 2.50% accuracy drop when removing the pretrained TSFM entirely (Table 6). This is a thin margin to support the paper's central thesis that pretrained temporal features are "crucial" for time series reasoning. The word "crucial" implies a large, clearly measurable contribution — the evidence suggests "helpful but modest."

2. No variance, standard deviation, confidence intervals, or results from multiple random seeds are reported across any experiment. This is the single most important gap. With improvements in the 2–4% range, I need to know these aren't just noise. I notice that Table 2 reports 54.83% OA for TS-Reasoner while Table 6 reports 54.26% for what appears to be the same model — this discrepancy (0.57%) itself hints at run-to-run variation that the paper doesn't discuss.

3. The evaluation is heavily weighted toward MCQA. While the paper does include an open-ended inductive reasoning task (Section 4.7), the primary benchmarks (TimeSeriesExam, MTBench) are both multiple-choice. MCQA has well-known issues: a 25% random baseline for 4-choice questions, position bias in LLMs, and the inability to distinguish genuine understanding from elimination strategies.

4. MTBench is an arXiv preprint (arXiv:2503.16858) that has not, to my knowledge, undergone full peer review. Using an unreviewed benchmark as one of two primary evaluation vehicles weakens the evidentiary foundation somewhat. This isn't disqualifying, but the paper should acknowledge the benchmark's status.

5. The paper does not sufficiently distinguish its contribution from concurrent 2025 work (ChronoSteer, TempoGPT, Time-VLM) that explores similar TSFM+LLM alignment. What specific design insight does TS-Reasoner contribute that these others don't?

6. The practical significance question remains under-explored. GPT-4.1 achieves 67.89% on TimeSeriesExam vs. TS-Reasoner's 54.83%. The paper doesn't clearly articulate scenarios where practitioners would prefer this approach over simply using a stronger LLM - cost, latency, privacy, domain adaptation potential are all plausible arguments, but the paper doesn't make them.

**Additional Comments:**

This is a competent piece of engineering that applies the established VLM paradigm to time series reasoning. The synthetic data pipeline and systematic ablations are the strongest elements. The paper reads well and is clearly structured.

A few non-critical suggestions: (1) Qualitative examples showing how attribute-aware captions differ from template captions in practice would strengthen Section 4.6 (the Appendix B examples are good but could be highlighted more). (2) Discussing which TimeSeriesExam categories show the largest gap vs. GPT-4.1 would surface informative failure modes. (3) The scaling analysis could discuss whether the TSFM's relative contribution changes with LLM capacity - does a bigger LLM compensate for the TSFM's absence?

With multi-seed experiments and appropriately scoped claims, this could be a solid TMLR contribution. The core ideas are sound; it's the evidentiary standard that needs work.

**Audience:**

Yes

**Audience Explanation:**

Clearly yes. Time series + LLM alignment is an active and growing area. Researchers working on multimodal foundation models, time series analysis, and synthetic data for low-resource modalities would all find useful takeaways here - the backbone comparison, data efficiency analysis, and captioning methodology are practically informative regardless of the statistical concerns above.

**Broader Impact Concerns:**

No major ethical concerns. The work uses synthetic data and public benchmarks, avoiding privacy issues. The time series reasoning capability has clear benign applications in finance and weather. The environmental cost of training (two-stage on 150K samples with 7B parameters) should be reported for transparency but is not unusual for this class of work.

**Claims And Evidence:**

No

**Claims Explanation:**

The directional evidence is promising, but the absence of statistical rigor makes it impossible to separate signal from noise at the margin sizes reported.

*Well-supported claims:*
- TS-Reasoner achieves higher accuracy than ChatTS-7B on TimeSeriesExam (54.83% vs. 50.72%) - clearly reported with specific numbers.
- Data efficiency: less than half the alignment data of ChatTS (120K vs. ~250K+) - straightforward factual comparison.
- TSFM backbone comparison (TimesFM > Chronos > MOMENT) - clear hierarchy with meaningful gaps, especially for MOMENT.
- Consistent improvement across LLM scales (0.5B, 3B, 7B) - Figure 6 shows TS-Reasoner leads at every scale.

*Weakly supported claims:*
- The pretrained TSFM provides "crucial" temporal representations: the 2.50% ablation gap without uncertainty quantification cannot substantiate "crucial." It may be real, but it may also be noise.
- Superiority over baselines: without multi-seed results, the 4.11% improvement over ChatTS-7B could reflect random variation. This isn't me being pedantic - the field has repeatedly seen reported improvements evaporate under proper statistical testing.

To change this assessment to "Yes": (1) report results across 3-5 random seeds with standard deviations, (2) conduct significance tests for key pairwise comparisons (paired bootstrap or McNemar's test would work), and (3) either provide stronger TSFM ablation evidence or recalibrate the claims to match the 2.5% reality.

**Requested Changes:**

**[Critical] Statistical robustness of main results**

All results are single-run values. With improvements of 2–4% in MCQA accuracy, I cannot distinguish real gains from noise. Please report results across at least 3 random seeds with standard deviations, and run significance tests (paired bootstrap or McNemar's) for the key comparisons: TS-Reasoner vs. ChatTS-7B, and the TSFM ablation.

The Table 2 vs. Table 6 discrepancy (54.83% vs. 54.26% for what should be the same model) makes this especially pressing — that unexplained 0.57% gap already suggests meaningful run-to-run variation.

**[Critical] Strengthen or recalibrate the TSFM contribution claim**

The paper's motivation centers on the frozen TSFM providing crucial temporal representations. A 2.50% ablation gap doesn't support "crucial." Two paths forward:

(a) Provide additional evidence: per-category breakdown showing where the TSFM helps most, representation probing experiments, qualitative analysis of what temporal patterns the TSFM captures that raw patches miss.

(b) Recalibrate the claims: frame the TSFM as "beneficial" rather than "crucial," and reposition the primary contribution around the synthetic data pipeline and training recipe, which are arguably the more distinctive elements anyway.

**[Important] Positioning against concurrent work**

Add a structured comparison (ideally a table) showing how TS-Reasoner differs from ChronoSteer, TempoGPT, and Time-VLM in terms of architecture, training data, frozen vs. fine-tuned components, and evaluation. What specific insight does this paper offer that the others don't?

**[Important] MTBench status acknowledgment**

Acknowledge that MTBench is a preprint without full peer review. Either add results on an established peer-reviewed benchmark, or clearly frame MTBench results as preliminary/supplementary rather than primary evidence.

**[Minor] Practical motivation**

Add a paragraph articulating when and why practitioners would use TS-Reasoner over GPT-4.1 (which scores 13 points higher). The open-source, lower-cost, privacy-preserving, domain-adaptable angles are all reasonable — just make them explicit.

**[Minor] Compute resources**

Report GPU type, number of GPUs, training time per stage, and total compute budget. This is standard for reproducibility and strengthens the data efficiency narrative.

---

> ### Author Response · Authors · 2026-06-14
> **Response to Reviewer fppB (Part I)**
>
> We thank Reviewer fppB for the thorough review and insightful suggestions. We address each concern as follows:
>
> __Q1__: Please report results across at least 3 random seeds with standard deviations, and run significance tests (paired bootstrap or McNemar's) for the key comparisons: TS-Reasoner vs. ChatTS-7B, and the TSFM ablation.
>
> __A1__: Thank you for your comments. Table 2 reports the accuracy averaged over three random seeds, we add standard deviations to the results:
>
> |Model|PR|NU|AD|SA|CA|OA|Finance (long)|Finance (short)|Weather (long)|Weather (short)|
> |:-|:-:|:-:|:-:|:-:|:-:|:-:|:-:|:-:|:-:|:-:|
> |TS-Reasoner|**53.46**±1.58|**60.15**±3.51|**53.23**±1.61|**63.42**±1.02|**43.39**±1.98|**54.83**±0.98|**92.00**±1.74|**93.28**±1.28|**60.44**±0.14|**61.55**±0.31|
>
> We conduct McNemar's test against both ChatTS-7B and our TSFM ablation. The results show that TS-Reasoner's improvements are statistically significant (p \< 0.05) on all subtasks except Similarity Analysis. This confirms that the gains over both ChatTS-7B and the TSFM-ablated variant are robust rather than attributable to variance.
>
> We indicate the significance test results in both Table 2 and Table 6.
>
> __Q2__: Strengthen or recalibrate the TSFM Provide additional evidence: per-category breakdown showing where the TSFM helps most.
>
> __A2__: Thank you for your insightful suggestion. In TS-Reasoner, TSFM plays a role in providing richer time series features for the backbone LLM, and instruction tuning enables the model to leverage such information for reasoning.
>
> We further analyze the benefits brought by TSFM:
>
> (1) By sequence length: we analyze the TSFM's effect with respect to time series length on MTBench. As shown below, the performance drop from removing the TSFM broadly increases with the sequence length, suggesting that the TSFM is increasingly beneficial as time series grow longer:
>
> |MTBench|Average time series length|TSFM Δ|
> |-|-|-|
> |Finance (short)|375|−2.15|
> |Weather (long)|336|−1.79|
> |Weather (short)|168|−0.62|
> |Finance (long)|135|−0.69|
>
> (2) By task type and domain: As shown in Table 6, removing the TSFM leads to a 2.5% overall drop on TimeSeriesExam but only a 1.3% drop on MTBench. We attribute this to their different reliance on time series. TimeSeriesExam questions depend almost entirely on the time series itself, so weaker temporal representations lead to larger performance drops. MTBench questions can also draw on contextual news/weather reports, which partially compensate for weaker temporal representations. This indicates that the TSFM is most beneficial when reasoning relies primarily on the time series, and its contribution becomes more modest when complementary textual context is available.
>
> We incorporate the above discussion and results into our Appendix D.1, and follow your suggestion to adjust the claim of TSFM in the main text.
>
> __Q3__: Positioning against concurrent work, including ChronoSteer, TempoGPT, Time-VLM.
>
> __A3__: Thank you for your helpful suggestion. We follow your suggestion to compare with these works as follows:
>
> | Method | Architecture | Training Data | Frozen / Fine-tuned | Evaluation tasks |
> |---|---|---|---|---|
> | ChronoSteer | LLM emits textual revision instructions to steer a TSFM's forecast (LLM→TSFM) | Synthetic instruction–series data | LLM and TSFM frozen, alignment module trained | Time series forecasting |
> | TempoGPT | VQ-VAE quantizes series into tokens; LLM vocabulary expanded with a shared embedding layer | Electrical time series simulation dataset | VQ-VAE frozen; LLM fine-tuned | Self-built electrical time series reasoning tasks |
> | Time-VLM | time series encoded as images via a pretrained VLM, fused for forecasting | Public time series forecasting datasets | VLM frozen, fusion network trained | time series forecasting |
> | TS-Reasoner (ours) | Frozen pretrained TSFM features projected into the LLM via an adapter for reasoning (TSFM→LLM) | Cross-domain captions + instructions | TSFM frozen; adapter + LLM fine-tuned | Public reasoning benchmarks + open-ended task |
>
> ChronoSteer's results indicate that the knowledge of LLM can be effectively integrated into TSFM to enhance the ability of time series prediction. Conversely, TS-Reasoner proves that the integration of TSFM knowledge into LLM improves the ability of LLM to understand time series. While TempoGPT tokenizes time series into discrete tokens via a quantization codebook, TS-Reasoner aligns the continuous embeddings of a pretrained TSFM with the LLM directly without any quantization, which avoids quantization-induced information loss. Time-VLM renders time series into images and pairs them with textual prompts to exploit a frozen VLM's alignment for time series forecasting, TS-Reasoner grounds the LLM in the continuous representations of a pretrained TSFM and shows that this is an effective route to time series understanding apart from visualization.
>
> We add this discussion in the Appendix C.

---

> ### Author Response · Authors · 2026-06-14
> **Response to Reviewer fppB (Part II)**
>
> __Q4__: The primary benchmarks (TimeSeriesExam, MTBench) are both multiple-choice. MCQA has well-known issues: a 25% random baseline for 4-choice questions, position bias in LLMs, and the inability to distinguish genuine understanding from elimination strategies.
>
> __A4__: Thank you for your comments. We agree that multiple-choice evaluation is a widely recognized yet still open challenge. TimeSeriesExam and MTBench are commonly adopted in recent time series reasoning work, and the multiple-choice formulation serves as a standardized interface enabling consistent comparison across models. In our setting, all baselines are evaluated under the identical protocol, so any format-induced bias applies uniformly and does not affect the relative improvements that constitute our central claims. Beyond multiple-choice, our results on the open-ended inductive reasoning task further suggest that TS-Reasoner generalizes to free-form evaluation.
>
> __Q5__: MTBench status acknowledgment.
>
> __A5__: Thank you for your comments. We acknowledge that MTBench is a preprint that has not undergone full peer review, and in the revision we explicitly revise the MTBench as supplementary evidence in Section 4.
>
> __Q6__: Practical motivation: Add a paragraph articulating when and why practitioners would use TS-Reasoner over GPT-4.1 (which scores 13 points higher). The open-source, lower-cost, privacy-preserving, domain-adaptable angles are all reasonable — just make them explicit.
>
> __A6__: Thank you for your suggestion. In practice, the choice between TS-Reasoner and proprietary models such as GPT-4.1 depends on deployment constraints. In domains such as finance and healthcare, data privacy regulations often prohibit sending time series to external cloud services, ruling out API-based proprietary models. As a fully open-source model that can be deployed locally, TS-Reasoner is well-suited to these settings, while requiring substantially lower computational cost than large proprietary multimodal models. This makes TS-Reasoner a practical choice when privacy, cost, or domain-specific adaptability as an important considerations. We add this discussion to a paragraph in Conclusion.
>
> __Q7__: Report GPU type, number of GPUs, training time per stage, and total compute budget.
>
> __A7__: Thank you for your suggestion. We use 8 $\times$ NVIDIA L40S GPUs. Stage-1 uses ~20 hours, and Stage-2 uses ~5 hours, in total 25 hours. We incorporate this detail in the paragraph Implementation Details in Section 4.
>
> __Q8__: (1) Qualitative examples showing how attribute-aware captions differ from template captions in practice would strengthen Section 4.6. (2) Discussing which TimeSeriesExam categories show the largest gap vs. GPT-4.1 would surface informative failure modes. (3) The scaling analysis could discuss whether the TSFM's relative contribution changes with LLM capacity - does a bigger LLM compensate for the TSFM's absence?
>
> __A8__: Thank you for your helpful suggestions. We respond your suggestions point-by-point as follows:
>
> (1) *Regarding the qualitative examples for difference between template captions and attribute-aware captions*: We follow your suggestion to add the comparison of template caption and attribute-aware captions in Figure 11. The caption generated by our attribute-aware captioning shows a greater language diversity and a more comprehensive, flexible description.
>
> (2) *Regarding the discussion of failure modes in TimeSeriesExam categories*: Among all categories, Pattern Recognition shows the largest gap to GPT-4.1. We attribute this primarily to the general multi-step reasoning capacity gap between our 7B backbone and a frontier-scale model. This suggests while TS-Reasoner improves the model's time series understanding, its overall performance on tasks that demand both temporal understanding and complex multi-step reasoning remains bounded by the reasoning capacity of the base LLM. A promising direction for future work is to further strengthen this reasoning ability on top of the current model to narrow the gap to frontier models. We add these into Section 4.1.
>
> (3) *Regarding the scaling analysis*: The results in Figure 6 show that the performance gain from incorporating the TSFM is markedly smaller at 0.5B than at 3B and 7B, indicating that the TSFM's contribution grows with model scale. We speculate that this is because smaller models have insufficient capacity and limited base reasoning ability, which constrains how well they can align with and exploit the TSFM features. In contrast, larger models possess a higher-dimensional, semantically richer representation space and stronger context-integration and multi-step reasoning abilities, allowing them to more fully align, interpret, and exploit the injected temporal features, thereby converting the fine-grained temporal information into substantial downstream gains. We add these discussions to Section 4.4 in the revision.

---

> > ### Comment · Reviewer_fppB · 2026-06-21
> > **I thank the authors for the thorough, point-by-point rebuttal and the carefully revised manuscript. I have read the full revision and checked each response against the changes in the paper. The revision substantively addresses my concerns**
> >
> > ### Resolved
> >
> > - **Statistical significance.** Table 2 now reports standard deviations and marks improvements that are significant under McNemar's test (p<0.05), and Table 6 carries the same marking against the "–TSFM" variant. The earlier Table 2 vs. Table 6 discrepancy (54.83 vs. 54.26) is also fixed — both now report 54.83. This directly addresses my primary concern about distinguishing signal from noise at the reported margins.
> > - **TSFM contribution claim.** I appreciate that the claim has been recalibrated from "crucial" to "beneficial," and that Appendix D.1 now characterizes *when* the TSFM helps most — by input length (Table 9) and by modality reliance (the larger drop on TimeSeriesExam than on MTBench). This is the honest, evidence-proportional framing I was asking for.
> > - **Positioning vs. concurrent work.** Appendix C and Table 8 clearly contrast TS-Reasoner with ChronoSteer, TempoGPT, and Time-VLM along architecture, training data, frozen/fine-tuned components, and evaluation, and articulate the key differentiators (continuous-embedding alignment, the TSFM→LLM direction, reasoning vs. forecasting).
> > - **Practical motivation.** The new Discussion paragraph in the Conclusion makes the privacy, cost, and domain-adaptability case for the approach.
> > - **Compute.** The Implementation Details now report 8×L40S GPUs and per-stage training time (~20 h + ~5 h).
> > - **Additional items.** The template vs. attribute-aware caption example (Figure 11), the largest-gap-to-GPT-4.1 analysis on Pattern Recognition (§4.1), and the discussion of how the TSFM's contribution scales with LLM size (§4.4) are all incorporated.
> >
> > I also note, and welcome, the 0% contamination analysis (Appendix F, Table 10) added in response to another reviewer — it speaks to the circular-evaluation risk I raised in my original review.
> >
> > ### Remaining minor items
> >
> > These are small and mostly textual; none requires new modeling work, and they do not block my positive assessment.
> >
> > 1. **Please state explicitly what the "±" values represent.** I could not find a sentence in the manuscript specifying the averaging protocol behind Table 2/Table 6. I also note that the response to me describes these as averaged "over three random seeds," whereas the response to Reviewer FzQt describes the same numbers as averaged "over three independent inference runs." These are not equivalent — the former characterizes training-run variance, the latter does not. Please reconcile the two and state the protocol clearly in the experimental setup. If the figures reflect inference-run variability over a single trained checkpoint, please say so and scope the robustness claim to McNemar's test-set significance accordingly. (A small multi-seed run on the headline configuration and the "–TSFM" ablation would fully settle this, if compute permits.)
> >
> > 2. **Please incorporate the same-pipeline base-LLM control into the paper.** The result reported to Reviewer FzQt — training Qwen2.5-7B-Instruct through the identical two-stage pipeline (time series as raw text) reaching 49.67 OA vs. TS-Reasoner's 54.83 — is, in my view, the most compelling single piece of evidence that the gains come from the time-series component rather than the additional data. It currently appears only in the discussion thread; please add it to the manuscript (e.g., as a row in Table 2 or in the ablation section).
> >
> > 3. **MTBench status.** Reframing MTBench as a "supplementary" benchmark is the right call. Please also add one short clause where the benchmark is introduced noting that it is a non-peer-reviewed preprint, so readers can weigh those results appropriately.
> >
> > 4. **Minor:** a one-line pointer from §2 to the new Appendix C comparison would aid discoverability.
> >
> > With these adjustments — particularly clarifying the variance protocol and folding in the same-pipeline control — I am satisfied that the central claims are supported by clear and appropriately scoped evidence. I thank the authors again for their responsiveness and the quality of the revision.

---

> > > ### Author Response · Authors · 2026-06-21
> > > **Thank you!**
> > >
> > > We sincerely thank Reviewer fppB for the positive evaluation and the detailed, valuable feedback. We address your remaining comments below.
> > >
> > > __Q1__: Please state explicitly what the "±" values represent.
> > >
> > > __A1__: Thank you for your comments. The protocol is as follows: three inference runs on a single trained checkpoint, each differing only in the random seed, and the ± values therefore denote standard deviations over these three runs and reflect inference-time variability. We add these details to the *Baselines and Evaluation Metrics* paragraph in Section 4 to state this protocol explicitly and to scope the robustness claim to McNemar's test-set significance.
> > >
> > > __Q2__: Please incorporate the same-pipeline base-LLM control into the paper.
> > >
> > > __A2__: Thank you for your suggestion. We include the same-pipeline training with base LLM in Table 6 and Section 4.8 Ablation study.
> > >
> > > __Q3__: Please also add one short clause where the benchmark is introduced noting that it is a non-peer-reviewed preprint, so readers can weigh those results appropriately.
> > >
> > > __A3__: Thank you for your suggestion. We explicitly point out MTBench is a non-peer-reviewed preprint in Paragraph *Datasets* in Section 4.
> > >
> > > __Q4__: Add an one-line pointer from §2 to the new Appendix C comparison would aid discoverability.
> > >
> > > __A4__: Thank you for your suggestion, we follow your suggestion to add a pointer in Section 2 to Appendix C for the comparison with different LLMs/VLMs architectures for time series.